# ReCon: Region-Controllable Data Augmentation with Rectification and Alignment for Object Detection

**Haowei Zhu[1], Tianxiang Pan[2], Rui Qin[1], Jun-Hai Yong[1], Bin Wang** [*1,3]
[1]Tsinghua University    [2]Li Auto Inc.    [3] BNRist
wangbins@tsinghua.edu.cn

## Abstract

The scale and quality of datasets are crucial for training robust perception models. However, obtaining large-scale annotated data is both costly and time-consuming. Generative models have emerged as a powerful tool for data augmentation by synthesizing samples that adhere to desired distributions. However, current generative approaches often rely on complex post-processing or extensive fine-tuning on massive datasets to achieve satisfactory results, and they remain prone to content–position mismatches and semantic leakage. To overcome these limitations, we introduce **ReCon**, a novel augmentation framework that enhances the capacity of structure-controllable generative models for object detection. ReCon integrates region-guided rectification into the diffusion sampling process, using feedback from a pre-trained perception model to rectify misgenerated regions within diffusion sampling process. We further propose region-aligned cross-attention to enforce spatial–semantic alignment between image regions and their textual cues, thereby improving both semantic consistency and overall image fidelity. Extensive experiments demonstrate that ReCon substantially improve the quality and trainability of generated data, achieving consistent performance gains across various datasets, backbone architectures, and data scales. Our code is available at https://github.com/haoweiz23/ReCon.

## 1   Introduction

Robust object detection and instance segmentation models are essential in modern computer vision (Bochkovskiy et al., 2020; Carion et al., 2020; Zhu et al., 2020; Liu et al., 2024). However, these models are highly dependent on large-scale, meticulously annotated datasets whose creation is expensive and time consuming (Barkai et al., 1993; Cherti et al., 2023). For instance, annotating a single image in the Cityscapes dataset can take up to 60 minutes (Cordts et al., 2016). Consequently, there is a pressing need for efficient and automated methods to synthesize high-quality annotated training data.

Data augmentation has emerged as a vital strategy to alleviate data scarcity by increasing sample diversity and improving model generalization. Traditional augmentation methods (Zhong et al., 2020; DeVries & Taylor, 2017; Yun et al., 2019; Cubuk et al., 2018; Dvornik et al., 2018) typically introduce only minor local variations, falling short of generating truly novel content. Recent advances in generative modeling, especially structurally controllable frameworks, offer a promising alternative by leveraging Canny edges (Zhang et al., 2023; Zavadski et al., 2024), spatial layouts (Chen et al., 2023; Wang et al., 2024b), or instance masks (Wang et al., 2024a; Wu et al., 2023) to maintain fine-grained control during image synthesis.

---

[*]Corresponding author.

39th Conference on Neural Information Processing Systems (NeurIPS 2025).

Structurally controllable generative models have achieved remarkable progress in geometric manipulation and are now extensively applied to object-detection data augmentation (Fang et al., 2024; Li et al., 2025). One family of methods repaints original images using universal control models such as ControlNet (Zhang et al., 2023; Zavadski et al., 2024) or inpainting models (Rombach et al., 2022; Lugmayr et al., 2022). These pipelines are often complex, requiring extra post-filters to remove noisy outputs (Fang et al., 2024) or multiple sampling processes (Kupyn & Rupprecht, 2024) to generate an image, for example, Kupyn & Rupprecht (2024) generating new samples for a single image containing ten objects via ten separate diffusion samplings.

Moreover, fine-tuning diffusion models conditioned on layouts or masks provides a feasible way for precise end-to-end synthetic data generation for object detection. Recent works demonstrate that such generated datasets yield strong trainability in downstream tasks (Chen et al., 2023; Wang et al., 2024b). However, these approaches typically require fine-tuning on large-scale datasets, which incurs significant computational overhead and remains impractical when data are scarce, which is a common scenario in data augmentation tasks. Furthermore, these approaches often struggle with complex layouts, leading to mis-generated regions and semantic misalignment.

To address these challenges, we propose **Region-Controllable (ReCon)** data augmentation. By integrating region-wise rectification and alignment directly into the diffusion sampling process, ReCon enhances single-pass control over instance synthesis. Without any additional training, ReCon significantly improves consistency between generated content and its annotations. It should be noted that we are not claiming to introduce a novel structural-control generation framework. Instead, our method can, without any additional training, enhance the quality of object detection data produced by existing structural controllable generation models. Specifically, our method first performs Region-Guided Rectification (RGR), in which we detect mis-generated regions by comparing the sampled image against ground-truth annotations using an off-the-shelf grounding model and then rectify those areas by injecting noisy real data points. By applying rectification to areas susceptible to be mis-generated, we boost the accuracy without sacrificing content diversity. Next, we introduce Region-Aligned Cross-Attention (RACA) to mitigate semantic leakage. This mechanism aligns region-specific visual tokens with their corresponding textual descriptions (or other cues) during generation. By enforcing a tight correspondence between image features and text embeddings at each diffusion step, it ensures precise semantic fidelity in the output.

By incorporating these two components into every sampling iteration, ReCon provides fine-grained region control: region-guided rectification preserves spatial agreement with annotations, and region-aligned cross-attention guarantees semantic adherence. The resulting high-fidelity augmented samples significantly enhance downstream object-detection performance. For example, when combined with a Canny-edge conditioned ControlNet model, our method achieves superior performance on the COCO dataset compared to models that have been specifically fine-tuned on COCO. In addition, our method demonstrates high augmentation efficiency: in data-scarce settings, tripling the dataset with our approach outperforms a sevenfold increase achieved by the baseline. Our contributions are as follows:

- We propose ReCon, a novel region-controllable data augmentation method that enhances the regional control capabilities of existing models without requiring additional training.

- We introduce region-guided rectification and region-aligned cross-attention mechanisms to improve control ability during the diffusion sampling process.

- Extensive experiments show that ReCon generates high-quality augmented data and substantially improves detection performance compared to both traditional augmentation techniques and current generative approaches.

## 2   Related Work

**Conditional Generation Models.** Recent advances in generative modeling have enabled the synthesis of high-fidelity images. Early research predominantly focused on Generative Adversarial Networks (GANs) (Goodfellow et al., 2020) for image generation. Conditional GANs, for example, were utilized to train classification heads, thereby demonstrating the capability of GANs to model data distributions and generate novel samples in an unsupervised manner (Gurumurthy et al., 2017; Antoniou et al., 2017; Mariani et al., 2018; Zhang et al., 2021; Li et al., 2022a; Zhao & Bilen, 2022;

Xu et al., 2023). However, GANs are often plagued by training instability, mode collapse, and limited controllability, particularly in low-data regimes.

Diffusion models have recently emerged as a robust alternative, offering enhanced controllability and adaptability. These models implement a reverse denoising process that gradually removes noise from an initial Gaussian distribution to approximate the real data distribution (Yang et al., 2023a). Moreover, diffusion models can effectively handle a variety of conditioning inputs, including text, images, layouts, edges, depth maps, points, and masks. This flexibility has enabled their application to a wide range of tasks such as text-to-image synthesis (Podell et al., 2023; Esser et al., 2024), image editing (Meng et al., 2021; Rombach et al., 2022), image inpainting (Lugmayr et al., 2022; Saharia et al., 2022), and data augmentation (Fang et al., 2024; He et al., 2022). For instance, LAMA (Li et al., 2021) proposed a large mask inpainting strategy to enhance image quality, while Taming Transformers (Esser et al., 2020) demonstrated that training in a latent space can outperform more complex baselines. Further innovations include GLIGEN (Li et al., 2023c), which incorporates gated self-attention for improved layout control, and LayoutDiffuse (Cheng et al., 2023), which employs layout-specific attention modules tailored for bounding box guidance. Additionally, methods like GeoDiffusion (Chen et al., 2023) and Instance Diffusion (Wang et al., 2024a) integrate geometry-aware modules to encode spatial features, leading to superior generation outcomes. DetDiffusion (Wang et al., 2024b) introduces a perception-aware loss to effectively bridge the gap between generation and perception.

In this paper, we exploit these advanced, controllable generative models to produce high-quality synthetic data without extra training, with the goal of enhancing downstream detection tasks.

**Generative Data Augmentation.** Recent advancements in generative models (Rombach et al., 2022; Esser et al., 2024; Tian et al., 2025) have paved the way for synthesizing high-fidelity images that introduce novel content beyond the capabilities of traditional augmentation techniques (Cubuk et al., 2020, 2018; Yun et al., 2019; Chen et al., 2020). This enhanced data diversity is instrumental in improving the training of perceptual networks for tasks such as object detection.

Initial studies employed GANs (Goodfellow et al., 2020) for data augmentation. However, subsequent research has revealed several limitations of GAN-based approaches. For example, training networks like ResNet50 (He et al., 2016) on data synthesized by models such as BigGAN (Brock et al., 2018) often results in suboptimal performance compared to training with real images. Moreover, the inherent instability in GAN training and the difficulty of generating data under complex conditions (Bansal & Grover, 2023; Gowal et al., 2021; Ravuri & Vinyals, 2019) further constrain their effectiveness.

In contrast, diffusion models offer superior controllability and have gained widespread application in data generation. For image classification, methods such as LECF (He et al., 2022) use GLIDE (Nichol et al., 2021) to generate images and subsequently filter out low-confidence samples to enhance zero-shot and few-shot performance. Similarly, SGID (Li et al., 2023a) leverages BLIP (Li et al., 2022b) to ensure semantic consistency in generated outputs. Feng et al. (2023) filters samples based on feature similarity, while techniques like GIF (Zhang et al., 2022) and DistDiff (Zhu et al., 2024) incorporate additional guidance during the sampling process to refine the quality of generated samples. For object detection, recent methods such as GeoDiffusion (Chen et al., 2023) and DetDiffusion (Wang et al., 2024b) have demonstrated the ability to synthesize high-quality images with precise layout control, specifically designed for training detection models. Additional strategies include using diffusion models with post-filtering based on category-calibrated CLIP scores (Fang et al., 2024) and applying background inpainting to augment training data without extra annotations (Li et al., 2025).

Moreover, synthetic data has shown promise in other domains as well. For instance, MagicDrive (Gao et al., 2023) highlights the benefits of synthetic samples for 3D perception tasks, while TrackDiffusion (Li et al., 2023b) focuses on data generation for multi-object tracking. X-Paste (Zhao et al., 2023) and MosaicFusion (Xie et al., 2024) further contribute by producing samples with clear segmentation boundaries to boost instance segmentation performance.

Despite these advances, most current methods either require additional training of generative models or struggle to balance fidelity and diversity. In this work, we develop the generative data augmentation framework for object detection by leveraging emerging zero-shot recognition models (e.g., GroundedSAM (Ren et al., 2024)) alongside versatile conditional generation models (e.g., Stable Diffusion (Rombach et al., 2022) and ControlNet (Zhang et al., 2023)). Our approach eliminates the

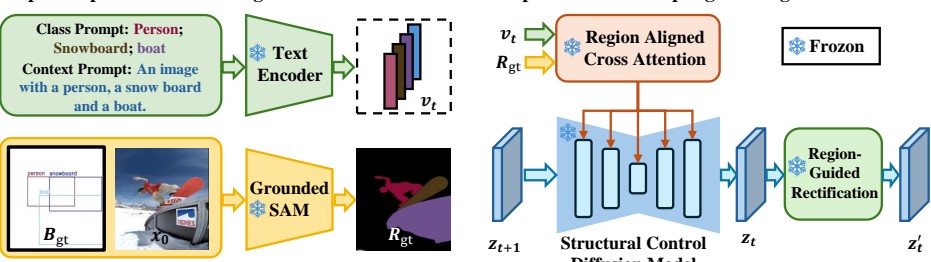

Figure 1: Overview of the ReCon Pipeline. ReCon enhances object-detection data generation by integrating region control into frozen, off-the-shelf models. It first compute the text embedding and instance masks, and then leverages a structural controllable diffusion model as the data generator and introduces region-guided rectification to refine generated results during the sampling process. Additionally, region-aligned cross-attention is incorporated to mitigate semantic leakage. Our method is plug-and-play and can be integrated with existing structure-controllable models.

need to retrain generative models, which is often impractical in data-scarce scenarios, and instead focuses on simplicity and effectiveness in generating task-specific data for target detectors.

## 3 Method

**Task Definition.** This study enhances a downstream object detector through data augmentation. By leveraging a pre-existing generator with the original image $x$, bounding boxes $B$, and class labels $y$, we aim to generate a high-fidelity augmented dataset where objects appear within the specified $B$ and carry the correct labels $y$. The primary challenge is to preserve fidelity to the source while introducing useful novel variations (e.g., new colors, styles, or object poses) to increase content diversity and thereby improve downstream model performance.

**Overview.** Existing methods often face a trade-off between diversity and fidelity in generating downstream data. Recent works improve data diversity by using in-painting techniques to preserve certain image regions while redrawing others (Li et al., 2025; Ma et al., 2024a). Others improve fidelity by using perceptual models like CLIP (Radford et al., 2021) to filter out low-confidence samples (Fang et al., 2024; Zhao et al., 2023). In this paper, we propose a novel approach that utilizes an off-the-shelf perceptual model to adaptively calibrate image content during sampling, achieving a better balance between diversity and fidelity.

As illustrated in Figure 1, our method builds upon existing structural control models (e.g., ControlNet) to establish an initial layout control. During the sampling process, we employ a region-guided rectification strategy that refines instance-level content by automatically filtering out erroneous or low-confidence samples. Additionally, we introduce a region-aligned cross-attention mechanism to facilitate effective interaction between image content and the corresponding textual features.

### 3.1 Preliminaries

Stable Diffusion is a generative model that synthesizes high-quality images from textual prompts by operating within a compressed latent space. It comprises forward and reverse diffusion stages.

**Forward Process.** In the forward process, noise is gradually added to the latent representation $\mathbf{z}_0$ of an image $x$, turning it into pure Gaussian noise $\mathbf{z}_T$ after $T$ timesteps. This diffusion process is modeled by:

$$q(\mathbf{z}_t \mid \mathbf{z}_{t-1}) = \mathcal{N}\Big(\mathbf{z}_t; \sqrt{\alpha_t}\,\mathbf{z}_{t-1},\, (1 - \alpha_t)\,\mathbf{I}\Big), \tag{1}$$

where $\alpha_t$ controls the balance between the previous latent state and the injected noise.

**Denoising Process.** Starting with $\mathbf{z}_T$ (pure Gaussian noise), the model iteratively predicts and removes noise to generate the clean latent $\mathbf{z}_0$. This reverse sampling process is modeled by:

$$p_\theta(\mathbf{z}_{t-1} \mid \mathbf{z}_t) = \mathcal{N}\Big(\mathbf{z}_{t-1}; \mu_\theta(\mathbf{z}_t, t), \Sigma_\theta(t)\Big), \tag{2}$$

where $\theta$ is the denoising network, $\mu_\theta(\mathbf{z}_t, t)$ and $\Sigma_\theta(t)$ denote the predicted mean and covariance. Sequentially applying this reverse process from $t = T$ down to $t = 1$ effectively removes the noise, recovering $\mathbf{z}_0$ for final image generation.

**Cross-Attention Mechanism.** In text-to-image generation, the text condition $\mathbf{v}_t$ (encoded by a text encoder like CLIP) is integrated into the latent space via cross-attention:

$$\text{Attention}(\mathbf{Q}, \mathbf{K}, \mathbf{V}) = \text{softmax}\left(\frac{\mathbf{Q}\mathbf{K}^\top}{\sqrt{d_k}}\right)\mathbf{V}, \tag{3}$$

where the query $\mathbf{Q}$ is derived from image features, and $\mathbf{K}$ and $\mathbf{V}$ derived from text embeddings. Here, $d_k$ denotes the key dimension. This mechanism injects semantic text information into the latent features at each denoising step, guiding the image generation to reflect the text prompt.

During sampling, noisy latent representations are progressively denoised while being continuously influenced by the text embeddings. Starting the denoising from different timesteps allows control over the editing intensity, balancing adherence to the original image content. However, since the Stable Diffusion model lacks inherent structural control, additional structure-controllable models are required for generating object detection data.

### 3.2 Region-Controllable Data Augmentation

**Structural Control with ControlNet.** ControlNet (Zhang et al., 2023) enhances diffusion models (e.g., Stable Diffusion) by conditioning them on structural cues such as edge, depth, or pose maps. It integrates trainable control layers as follows:

$$\hat{\mathbf{z}}_l = \mathbf{z}_l + \gamma \cdot \text{ControlBlock}(\mathbf{c}_m, \theta_c) \tag{4}$$

where $\mathbf{z}_l$ are the latent features at layer $l$, $\mathbf{c}_m$ is the structural conditioning map, $\theta_c$ are learnable parameters of control blocks, and $\gamma$ scales the control signal.

In our work, we use ControlNet with an edge canny map to enforce structural constraints during image generation, and we demonstrate that our approach can generalize to other layout-to-image models for diverse guided generation.

**Region-Guided Rectification.** Existing generative models often encounter issues such as generating an incorrect number of target objects or unintended ones. These challenges significantly affect the quality of the generated data. To address these problems, we propose a region-guided rectification method aimed at perceiving image content during sampling and applying region adjustments. This approach ensures consistency of the content and the layout.

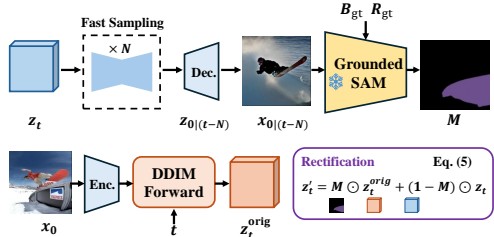

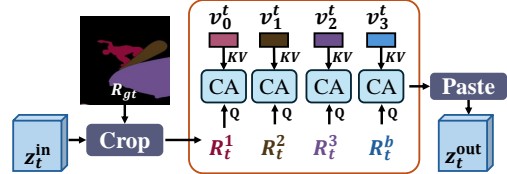

Figure 2: Sampling process with Region-Guided Rectification . We first identify incorrectly generated regions by IoU matching detection results with the original annotations using a grounding model. Next, we derive a rectification mask $\mathbf{M}$ and use $\mathbf{z}_t^{orig}$ sampled from the original data point to correct these errors.

Figure 3: Pipeline of our Region-Aligned Cross-Attention. We first crop region-specific features from $\mathbf{z}_t^{in}$ using predefined regions $R_{gt}$. For regions belonging to the same category, we perform cross-attention with the corresponding text features. Finally, the interacted region image features are concatenated to produce $\mathbf{z}_t^{out}$.

As shown in Figure 2, given an image annotation comprising multiple bounding boxes $B$ and their corresponding labels $y$, we employ the Grounded-SAM model (Ren et al., 2024) to detect potential objects in the data point $\mathbf{z}_t$ during the sampling process. We then apply IoU-based matching to identify false positives and false negatives, allowing us to segment regions that are potentially misgenerated.

These out-of-control regions are defined by a binary mask $\mathbf{M}$. The identified regions are then replaced with their corresponding noised versions $\mathbf{z}_t^{\text{orig}}$ sampled from the original image, while the remaining parts of $\mathbf{z}_t$ are preserved. This region-guided rectification process can be formulated as:

$$\mathbf{z}_t' = \mathbf{M} \odot \mathbf{z}_t^{\text{orig}} + (1 - \mathbf{M}) \odot \mathbf{z}_t, \tag{5}$$

where $\odot$ denotes the element-wise multiplication, $\mathbf{z}_t'$ represents the rectified latent data point, and $\mathbf{z}_t^{\text{orig}}$ is the latent point obtained after applying $t$ steps of noise addition to the original image using Equation 1. This method leverages the intrinsic *overridability* property of diffusion sampling (Levin & Fried, 2023), allowing regions in intermediate images to be replaced with external content drawn from the same distribution. Such rectifications can influence the final generated image without disrupting the overall inference process.

However, directly detecting the intermediate latent point $\mathbf{z}_t$ with a perception model is impractical due to the ***challenge of finding a pre-trained detector that provides meaningful guidance when the input is noisy***. To address this issue, we leverage recent cache-based diffusion acceleration method (Ma et al., 2024b) to speed up sampling over $N$ steps (with $N = 5$ by default). Furthermore, we utilize the capability of the diffusion model to predict the noise added to $\mathbf{z}_{t-N}$, enabling the prediction of a clean data point $\mathbf{z}_{0|t-N}$. This process is formulated in Equation 6.

$$\mathbf{z}_{0|t-N} = \frac{\mathbf{z}_{t-N} - \sqrt{1 - \alpha_t}\theta(\mathbf{z}_{t-N}, t)}{\sqrt{\alpha_t}}, \tag{6}$$

Then we apply region rectification at $T_r$ timesteps ($T_r = 4$), corresponding to the early (0.75 $T$), middle (0.50 $T$), latter (0.25 $T$) and final (0.10 $T$) stages of the diffusion process. At the early stage, when the overall object layout has begun to emerge, we can correct inaccuracies in the spatial distribution of objects. During the middle stage, as the object starts to take shape, we rectify any incorrect semantic content in the objects. Finally, in the latter and final stages, we refine regions with suboptimal generation quality. More details of the sampling process are presented in Algorithm 1.

**Region-Aligned Cross Attention.** In text-to-image generation, semantic leakage often occurs, where the content of target regions does not align with the actual textual descriptions. To address this issue, we introduce region-aligned cross attention to mitigate information leakage across regions.

Since attention within the text encoder operates on all prompt tokens, and these tokens may belong to different categories, interference between category-specific features can occur (see the appendix Figure 8). To address this, we individually encode $C$ textual features for $C$ target categories using prompts in the format: [CLASS], as shown in Figure 1. Additionally, we employ a global context description to represent the overall scene, which interacts with the background region. For datasets like COCO, we can directly utilize the provided caption annotations or simply use a custom prompt, such as: "An image with two cars and three persons".

Next, as presented in Figure 3, we perform cross-attention interactions between the corresponding object regions and their associated textual features. This step alleviates information leakage in prompt descriptions by ensuring that region-specific textual features influence only their respective regions. An alternative approach to implementing region-aware cross-attention is to use an cross attention mask to suppress text features from unrelated categories, as proposed in (Xue et al., 2023). However, we have observed that, due to the lack of disentanglement during the encoding of textual features from different categories, the masked attention mechanism still suffers from semantic leakage. Besides, Instance Diffusion introduces an instance-masked attention and fusion mechanism to integrate region-specific conditions with corresponding visual tokens. However, it relies on additional region-specific modules and requires retraining to achieve satisfactory performance. In contrast, our approach mitigates the problem of semantic leakage and enhances the fidelity of generated images without the need for additional fine-tuning. Furthermore, we demonstrate that our method can be effectively combined with Instance Diffusion to further boost its performance, as shown in Table 1 and Figure 8.

## 4 Experiments

### 4.1 Experiment Settings

We evaluate our method by augmenting downstream object detectors with synthetic samples. We use Stable Diffusion v1.5 (Rombach et al., 2022) with a 25-step DDIM sampler (Song et al., 2020) and

edge-conditioned ControlNet (Zhang et al., 2023) to generate training images. These samples are combined with the original trainset and used to jointly train object detectors. We implement training and evaluation code based on the MMDetection framework (Chen et al., 2019). For consistency with prior work (Wang et al., 2024b), our default detector is Faster R-CNN (Ren et al., 2015) with an R-50-FPN backbone trained for six epochs. We also evaluate our method with diverse detectors including RetinaNet (Lin et al., 2017), ATSS (Zhang et al., 2020), FCOS (Tian et al., 2019), YOLO-X (Ge et al., 2021), and DEIM (Huang et al., 2024). Following previous works, we select images containing 3 to 8 objects for data generation, resulting data set comprising 47,200 images with 227,406 objects. For VOC benchmark, we combine the training sets of VOC 2007 and VOC 2012 for model training, with evaluation performed on the VOC 2007 test set (4,952 images). We use mAP (mean Average Precision), mAR (mean Average Recall), FID to evaluate the performance. Extensive experiments are conducted across various datasets, backbone architectures, and data scales.

## 4.2 Main Results

**Compared with State-of-the-art Methods.** We compare our approach with state-of-the-art structure-controllable generative diffusion models, as summarized in Table 1. The results demonstrate that our method significantly enhances the effectiveness of structure-guided techniques for object detection data augmentation. Specifically, we evaluate both general-purpose control methods (e.g., ControlNet) and models fine-tuned on COCO (e.g., DetDiffusion). When combined with these methods, our approach further improves their performance and establishes a new state of the art. For instance, integrating ReCon with ControlNet yields a mAP of 35.5, surpassing GeoDiffusion's 34.8. Moreover, our method can act as a plug-and-play enhancement for region-controlled diffusion models without requiring additional training. This is exemplified by the improvement in GLIGEN's mAP from 34.6 to 35.5. These findings validate that our approach enables the generation of higher-quality training samples, resulting in a substantial boost in object detection performance.

Table 1: Comparison with existing generative models on the COCO dataset. ReCon enhances the detector performance by integrating existing methods in a training-free manner. The best results are highlighted in **bold**, while the second-best outcomes are denoted by _underlined italic_.

| Method | mAP | $AP_{50}$ | $AP_{75}$ | $AP^m$ | $AP^l$ |
|---|---|---|---|---|---|
| Real only | 34.5 | 55.5 | 37.1 | 37.9 | 44.3 |
| ▷ _General Control_ | | | | | |
| Layout Diffusion (Zheng et al., 2023) [CVPR23] | 34.0 | 54.5 | 36.5 | 37.2 | 43.6 |
| ControlNet (Zhang et al., 2023) [ICCV23] | 34.9 | 55.5 | 37.7 | 38.2 | 45.5 |
| Background-inpainting (Li et al., 2025) [ECCV24] | 35.1 | 55.1 | 37.7 | 38.2 | 45.8 |
| ControlNet-XS (Zavadski et al., 2024) [ECCV24] | 35.1 | 55.8 | 37.6 | 38.6 | 45.0 |
| ▷ _Fine tuned on COCO_ | | | | | |
| ReCo (Yang et al., 2023b) [CVPR23] | 33.6 | 53.2 | 36.2 | 36.7 | 44.0 |
| GLIGEN (Li et al., 2023c) [CVPR23] | 34.6 | 55.1 | 37.2 | 38.1 | 44.7 |
| GeoDiffusion (Chen et al., 2023) [ICLR24] | 34.8 | 55.3 | 37.4 | 38.2 | 45.4 |
| DetDiffusion (Wang et al., 2024b) [CVPR24] | 35.4 | 55.8 | 38.3 | 38.5 | **46.6** |
| Instance Diffusion (Wang et al., 2024a) [CVPR24] | 35.0 | 55.4 | 37.6 | 38.4 | 45.7 |
| ControlNet + **ReCon** | _35.5_ | **56.2** | **38.4** | **39.0** | 46.0 |
| GLIGEN + **ReCon** | 35.3 | _56.0_ | 38.1 | 38.7 | 45.8 |
| Instance Diffusion + **ReCon** | **35.6** | 56.0 | _38.4_ | _39.0_ | _46.4_ |

**Data-Scarce Scenarios.** Data augmentation is crucial when training data is limited. To evaluate our approach under such conditions, we conduct experiments in three data-scarce regimes by randomly sampling 1%, 5%, and 10% of the COCO training set and then doubling each subset through augmentation. Our method delivers consistent gains over baseline approaches in all regimes. As shown in Table 2, with only 10% of the data, mAP rises from 18.5% to 21.7%. Training-based generative models often struggle in data-scarce settings due to their dependence on large datasets. In contrast, we employ a generic structure-controlled diffusion model (ControlNet) to produce high-quality object detection samples. We further compare our method to traditional augmentation method RandAugment (Cubuk et al., 2020). Although RandAugment shows noticeable improvement, it

remains inferior to our approach. Moreover, combining our method with RandAugment produces additional improvements, demonstrating compatibility with standard augmentation pipelines.

**Few-Shot Scenarios.** We also evaluated our method in a 30-shot training setting on YOLOX-S (Ge et al., 2021) using COCO dataset, following the few-shot split protocol of previous work (Wang et al., 2020). Our method performs well even under the few-shot setting, increasing the mAP from 5.4 to 6.7 and $AP_{50}$ from 10.3 to 12.3. More few-shot results are presented in Table 11 in Appendix.

Table 2: Comparison in data-scarce scenarios across varying data proportions.

| Method | 1% | 5% | 10% |
|---|---|---|---|
| Real only | 0.3 | 13.0 | 18.5 |
| RandAugment | 3.1 | 16.2 | 21.4 |
| ControlNet | 2.5 | 15.9 | 21.2 |
| **Recon** | 3.9 | 16.7 | 21.7 |
| **Recon** + RandAugment | **4.2** | **17.1** | **22.0** |

**Comparison on Different Dataset.** To validate the generalization capability of our method, we conducted additional experiments on PASCAL VOC datasets. We compared our approach against a baseline Faster R-CNN detector trained with $1\times$ schedule. As demonstrated in Table 3, traditional data augmentation methods like RandAugment (Cubuk et al., 2020) show limited effectiveness, while simply duplicate the original dataset leads to overfitting (76.2 vs. 77.1 mAP). Our method achieves superior performance (78.5 mAP) through synthetic generation of diverse high-fidelity images that maintain crucial semantic features.

Table 3: Comparison results on the PASCAL VOC dataset.

| Method | Real only | Simple Duplicate | RandAugment | ControlNet | **ReCon** |
|---|---|---|---|---|---|
| mAP | 77.1 | 76.2 | 77.7 | 77.8 | **78.5** |

**Data Scaling.** To assess scalability we measure detection accuracy in low-data regimes (5% and 10%), summarized in Figure 4. Repeating training data (real expansion) provides consistent improvements up to $3\times$: mAP increases from 13.0 to 17.1 on the 5% subset and from 18.5 to 21.1 on the 10% subset. However, further duplication ($5\times$, $7\times$) leads to saturated performance and noticeable degradation, indicating overfitting under extended training. By contrast, our method generates diverse, annotation-consistent samples and continues to yield

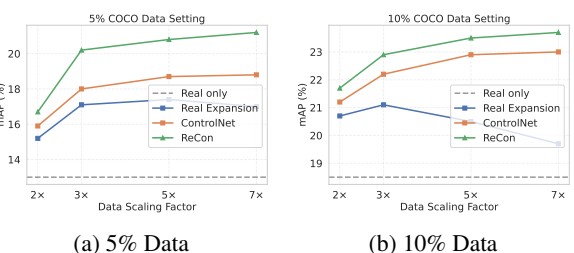

(a) 5% Data      (b) 10% Data

Figure 4: Data scaling on different COCO subsets.

performance gains without overfitting. As the expansion scale grows, the relative accuracy improvements from our augmented data become more pronounced. Overall, our approach attains comparable or better performance than the ControlNet baseline using substantially fewer augmented examples, highlighting its efficiency for data augmentation.

## 4.3 Ablation Studies

**Effectiveness of Each Component.** We conducted ablation experiments to assess the contributions of each component in our proposed framework, as presented in Table 4. The results clearly indicate that each component enhances the performance of the downstream model. In particular, the integration of region-guided rectification (RGR) and region-aligned cross attention (RACA) significantly improves the consistency between the generated samples and their corresponding annotations, thereby elevating the quality of the synthesized data. Consequently, our approach increases the baseline mAP from 34.9 to 35.5 and improves the FID from 13.82 to 12.85, demonstrating enhanced trainability and fidelity of the generated samples.

Table 4: Ablation results for different components of our proposed method.

| RGR | RACA | FID | mAP | AP$_{50}$ | AP$_{75}$ | AP$^m$ | AP$^l$ |
|---|---|---|---|---|---|---|---|
| ✗ | ✗ | 13.82 | 34.9 | 55.5 | 37.7 | 38.2 | 45.5 |
| ✔ | ✗ | 13.21 | 35.3 | 56.0 | 38.1 | 38.6 | 45.6 |
| ✔ | ✔ | **12.85** | **35.5** | **56.2** | **38.4** | **39.0** | **46.0** |

Table 5: Performance comparison of different perception targets.

| Target | mAP | AP$_{50}$ | AP$_{75}$ |
|---|---|---|---|
| $x_t$ | 35.0 | 55.6 | 37.8 |
| $x_{0|t}$ | 35.3 | 55.8 | 38.2 |
| $x_{0|(t-N)}$ | **35.5** | **56.2** | **38.4** |

Table 6: Trainability comparison with DEIM-D-FINE-N (Huang et al., 2024) on COCO.

| Method | mAP | AP$_{50}$ | AP$_{75}$ | mAR |
|---|---|---|---|---|
| Real only | 38.5 | 55.2 | 41.5 | 60.4 |
| ControlNet | 39.1 | 55.8 | 42.1 | 60.6 |
| **ReCon** | **39.8** | **56.6** | **42.5** | **61.0** |

Table 7: Comparison with different region-guided models.

| Method | mAP | AP$_{50}$ | AP$_{75}$ |
|---|---|---|---|
| Swin-Tiny | 35.5 | 56.2 | 38.4 |
| Swin-Base | **35.6** | **56.2** | **38.7** |

**Perception Target.** Our method leverages cache-based fast sampling (Ma et al., 2024b) to recover a clean $x_0$, providing a more accurate control signal for region-guided rectification. we compare different perception targets: $x_t$, $x_{0|t}$, and $x_{0|(t-N)}$. As shown in Table 5. While $x_t$ yields modest gains due to low recall which in turn causes the model to favor a lower overall editing strength. In contrast, employing $x_{0|t}$ further enhances performance, and the best results are achieved when using $x_{0|(t-N)}$ obtained via the fast sampling method.

**Different Detection Backbone.** We evaluate multiple object detectors and report results on the state-of-the-art DEIM (CVPR25) method in Table 6. Additional detector comparisons are provided in Table 10 in the Appendix. Our experiments show that our method consistently improve performance across different detectors, demonstrating its robustness and effectiveness.

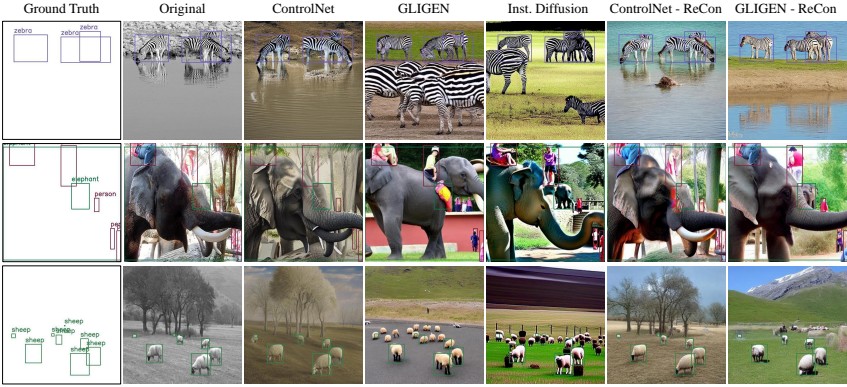

Figure 5: Visualization comparison of generated samples. Our methods show better image fidelity and content-annotation consistency.

**Perception Model.** Our approach employs the Grounded-SAM (Ren et al., 2024) as region-guided model to provide region-aware guidance. Additionally, we compare different backbone models for the detector within Grounded-SAM, as detailed in Table 7. The experimental results indicate that better perception leads to improved performance, suggesting that our method stands to benefit from stronger foundation models.

### 4.4 Qualitative Results

Figure 5 shows that our method substantially improves both the fidelity and the localization accuracy of generated samples. Unlike prior structure-control methods such as GLIGEN and ControlNet, which lack mechanisms for fine-grained region rectification and hence can exhibit imprecise localization and semantic leakage, our approach enforces strict consistency with the provided annotations. For

example, it removes an extraneous zebra produced by GLIGEN outside the target bounding box (row 1) and a superfluous sheep outside the region of interest (row 3), and it correctly restores a person that ControlNet fails to generate (row 2). By aligning generated content with the original annotations, our method improves overall generation quality while maintaining high fidelity and sample diversity. More visualization results are provided in the Appendix.

## 5   Conclusion

This paper presents **Re**gion-**Con**trollable data augmentation (ReCon), a training-free, diffusion-based method developed to generate high-quality, content-label-aligned synthetic data for enhancing object detection models. Extensive experimental evaluations demonstrate that ReCon outperforms traditional augmentation and generative methods, ultimately leading to superior detection performance.

## 6   Acknowledgments

This work was supported by the National Natural Science Foundation of China under Grant 62021002.

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

# A    Limitations and Societal Impacts

**Limitations.**    Although ReCon produces better FID scores and improves detector mAP without additional training, it may increase computation time as data volume grows. The requirement for an extra perception model also raises development costs. We have introduced acceleration techniques to lower these costs. Integrating a fast sampler (Luo et al., 2023) and a lightweight perception model offers a promising path to maximize the efficiency of diffusion based data augmentation for detector training.

**Societal Impacts.**    Generative models (Podell et al., 2023; Zavadski et al., 2024) offer a cost-effective alternative to manual data collection and annotation for object detection. Our method enhances these models' ability to produce annotation-consistent content without additional training and improves downstream detector performance. This efficiency can benefit organizations and researchers who face limited data resources.

However, because generative models are pre-trained on large, uncurated vision–language datasets from the internet, they may inherit social biases and stereotypes (Cho et al., 2023; Naik & Nushi, 2023; Ross et al., 2020) and produce discriminatory outputs. It is therefore essential to integrate bias detection and mitigation mechanisms. By applying region-wise rectification and alignment, our approach improves content accuracy and reduces bias.

Another concern is the potential misuse of synthetic imagery for purposes such as deepfakes (Lyu, 2020), which can spread misinformation and undermine societal trust. To address this risk, the community must establish regulations and best practices that ensure responsible creation and use of synthetic data models.

# B    Pseudo Algorithm

We present the pseudo algorithm of our Region-Guided Rectification Sampling in Algorithm 1. In **Stage 1**, we use a pre-trained perception model to identify instance masks. If annotated masks are already available, this step can be skipped. Next, we apply *exclusive dilation* to mitigate boundary segmentation issues and obtain a refined mask region $R_{gt}$, which serves as the initial ground-truth region. If visual priors such as Canny edge maps are provided, we further suppress false-positive regions in these priors, as they are prone to misidentification and may introduce noise in the corresponding visual condition maps.

In **Stage 2**, we define a rectification interval of $T_r$ steps within the sampling process. First, we perform $N$ steps of fast sampling to infer the latent $z'_{0|t-N}$, which is then decoded into an image. This image is passed through an object detector to identify false positives and false negatives. We then segment the corresponding regions: false positives are segmented on $x'_{t-N}$, and false negatives on the reconstructed image $x_0$. These regions are merged to produce a region-guided correction mask $M$. To rectify errors, we generate a noisy latent $z_t^{\text{orig}}$ by adding $t$ steps of noise to the original image. Finally, we mix $z_t$ and $z_t^{\text{orig}}$ using the region-guided mask to correct misgenerated areas.

# C    Experimental Setting

## C.1    Metrics Definition

We provide a detailed explanation of the metrics used to evaluate the effectiveness of our method below:

**mAP:** The mean of the average precision values computed over multiple IoU thresholds (typically from 0.50 to 0.95 in steps of 0.05) and across all classes, reflecting overall detection accuracy under varying overlap requirements.

**AP$_{50}$** : The average precision at a single IoU threshold of 0.50, measuring detection quality under a more lenient overlap criterion.

**AP$_{75}$:** The average precision at a single IoU threshold of 0.75, measuring detection quality under a stricter overlap criterion.

---

**Algorithm 1** Region-Guided Rectification Sampling Process

---

**Require:** VAE decoder $\theta_d$, Object detector $\mathcal{D}$, segmentation model $\mathcal{S}$
**Require:** Original image $x_0$, initial latent $z_0$, control map $c$, text prompt $p$, initial noise $\epsilon$
**Require:** Ground-truth boxes $B_{\text{gt}}$ with labels $y_{\text{gt}}$
**Require:** Refinement time steps $T_r$, fast-sampling step count $N$
**Ensure:** Final output image $\hat{x}_0$

1: **Stage 1: Initial Mask preparation**
2:   $B_{\text{pred}}, y_{\text{pred}} \leftarrow \mathcal{D}(x_0)$          ▷ Detect candidate boxes in original image
3:   $B_{fp} \leftarrow \text{match\_by\_IoU}(B_{\text{pred}}, B_{\text{gt}}, y_{\text{pred}}, y_{\text{gt}}, \tau = 0.5)$          ▷ Select false positives
4:   $(R_{\text{gt}}, R_{fp}) \leftarrow \mathcal{S}(x_0, [B_{\text{gt}}, B_{fp}])$          ▷ Segment GT and FP regions
5:   $(R_{\text{gt}}, R_{fp}) \leftarrow \text{ExclusiveDilate}(R_{\text{gt}}, R_{fp}, \text{kernel} = 7)$      ▷ Dilate masks and avoid overlap
6:   $c \leftarrow c \odot (1 - R_{fp})$          ▷ Keep control without FP regions
7: **Stage 2: Sampling with region-guided rectification**
8: **for** $t = T - 1, T - 2, \ldots, 0$ **do**
9:     $z_t = \sqrt{\alpha_t} \dfrac{z_{t+1} - \sqrt{1 - \alpha_{t+1}}\, \epsilon_\theta(z_{t+1}, t+1)}{\sqrt{\alpha_{t+1}}} + \sqrt{1 - \alpha_t}\, \epsilon_\theta(z_{t+1}, t+1)$      ▷ Denoise
10:     **if** $t \in T_r$ **then**
11:         $z'_{t-M} \leftarrow \text{FastSample}(z_t, N)$          ▷ Accelerated $N$-step sampling
12:         $z'_{0 \mid t-N} \leftarrow \dfrac{z'_{t-N} - \sqrt{1 - \bar{\alpha}_t}\, \epsilon_\theta(z_t, t)}{\sqrt{\bar{\alpha}_t}}$
13:         $x'_{0 \mid t-N} \leftarrow \theta_d(z'_{0 \mid t-N})$
14:         $B_{\text{pred}}, y_{\text{pred}} \leftarrow \mathcal{D}(x'_{0 \mid t-N})$
15:         $(B_{fp}, B_{fn}) \leftarrow \text{match\_by\_IoU}(B_{\text{pred}}, B_{\text{gt}}, y_{\text{pred}}, y_{\text{gt}}, \tau = 0.5)$      ▷ Select false
    positives and false negatives bboxes
16:         $M \leftarrow \text{merge}(\mathcal{S}([x'_{0 \mid t-N}, x_0], [B_{fp}, B_{fn}]))$    ▷ Merge false positive and false negative
    region to build region-rectification mask
17:         $z_t^{\text{orig}} = \sqrt{\alpha_t} z_0 + \sqrt{1 - \alpha_t} \epsilon$
18:         $z_t \leftarrow z_{t \mid 0} \odot M + z_t \odot (1 - M)$          ▷ Region-Guided Rectification
19:     **end if**
20: **end for**
21: $\hat{x}_0 \leftarrow \theta_d(\hat{z}_0)$
22: **return** $\hat{x}_0$

---

$\text{AP}^m$ : The average precision for medium-sized objects (area $\in$ [32², 96²] pixels), indicating performance on objects of moderate scale.

$\text{AP}^l$ : The average precision for large objects (area > 96² pixels), indicating performance on larger targets that are generally easier to localize.

**FID**: Fréchet Inception Distance (FID) measures the similarity between real and generated images. We compute FID using 5,000 samples. Lower FID indicates higher image quality and diversity.

## C.2 More Training Details

In this study, we train several object detection models on downstream detection datasets, including Faster R-CNN (Ren et al., 2015) with an R50 FPN backbone, ATSS (Zhang et al., 2020) with an R50 FPN backbone, FCOS (Tian et al., 2019) with an R50 FPN backbone, YOLOX-S (Ge et al., 2021), RetinaNet with a Swin-Tiny (Liu et al., 2021) FPN backbone and DEIM-D-FINE -N (Huang et al., 2024) model. For Faster R-CNN, ATSS, FCOS, and RetinaNet, we follow the standard 1× training schedule, running 12 epochs for all experiments, except for Faster R-CNN on the COCO dataset, where the training is reduced to 6 epochs. We use random flipping as the default data augmentation strategy. For YOLOX-S, we follow the official training setup with 300 epochs and apply a stronger augmentation pipeline, including mosaic, random affine transformations, mixup, random flipping, and HSV-based random augmentation. For DEIM, we follow the official training configuration to train model for 40 epochs.

# D    More Visualization Results

## D.1    Visualization Samples Analysis

As shown in Figure 6, we present additional samples generated by our method, which illustrate its ability to produce novel content with high image-annotation consistency and strong visual fidelity. Our method remains robust even in challenging scenarios involving tiny object boxes and densely distributed objects, ultimately improving the effectiveness of downstream detection models.

In Figure 6, we notice that some of the smaller objects in our generated images remain strikingly similar to those in the originals (the plant in the second row). This is partly due to the Canny-conditioned ControlNet, which helps preserve source-like structures, a result we see much less often with GLIGEN. More fundamentally, when a region proves difficult to synthesize accurately, our method continuously applies region-guided corrections to bring it closer to the original image. We lower the editing intensity in these challenging areas in order to maintain their structural and semantic integrity, so that any differences from the source appear only in texture and lighting.

Texture-level perturbations on hard-to-generate samples also help us produce valuable corner cases. For example, Figure 7 shows a truck on the left that is heavily occluded and has an implausible aspect ratio. By introducing controlled texture disturbances into the real image, we can create new corner cases that further enhance the model's robustness.

Additionally, a case study is provided in Figure 8, which demonstrates that although Instance Diffusion incorporates instance-level descriptions with specific visual tokens, it still suffers from semantic leakage. Our method can further mitigate this issue and enhance performance in such cases.

## D.2    Rectification Mask Analysis

Our method introduces region-guided rectification to correct misgenerated areas during the diffusion sampling process. We quantify the rectification mask's area at different sampling stages to analyze how much of the image each correction step influences. Across 1,000 samples, we measured the ratio of mask area to total image area at the 75%, 50%, 25%, and 10% sampling steps, obtaining 12.16%, 8.87%, 7.12%, and 6.77%, respectively. This trend demonstrates that our rectification procedure progressively reduces misgenerated regions as sampling proceeds.

# E    More Experimental Results

## E.1    Comparison of Rectification at Different Diffusion Timesteps

We study the sensitivity of our region-rectification schedule to the choice of diffusion timesteps. Concretely, we apply rectification at single timesteps as well as at several multi-stage schedules and report mean average precision (mAP), $AP_{50}$ and $AP_{75}$ in Table 8. Applying rectification only at an early stage ($0.75T$) produces a modest improvement over the ControlNet baseline (mAP 35.2 vs. 34.9), but is not optimal. We attribute this to two factors: (1) early-stage rectification can create shortcuts that allow spurious regions to be propagated and "fixed" during later synthesis steps, and (2) features at $0.75T$ remain highly diverse and only partially developed, so a single early rectification cannot fully suppress all errors.

When rectification is applied at multiple stages we observe consistent improvements. A two-stage schedule $[0.5T, 0.25T]$ raises mAP to 35.3, and a three-stage schedule $[0.5T, 0.25T, 0.1T]$ further increases mAP to 35.4. Our four-stage schedule $[0.75T, 0.5T, 0.25T, 0.1T]$ achieves the best balance across metrics (mAP 35.5, $AP_{50}$ 56.2, $AP_{75}$ 38.4). Extending the schedule to six stages yields only marginal additional benefit (mAP 35.5, $AP_{75}$ 38.5), indicating that performance gains saturate beyond four rectification stages. These results justify our chosen four-stage schedule as an effective and parsimonious design choice.

## E.2    Comparison with Image Editing Method

Existing methods suffer from semantic leakage and discrepancies between generated content and the original annotations. To address these issues, our approach calibrates intermediate sampling results using latent point sampled within the diffusion sampling process. We also explore the effectiveness

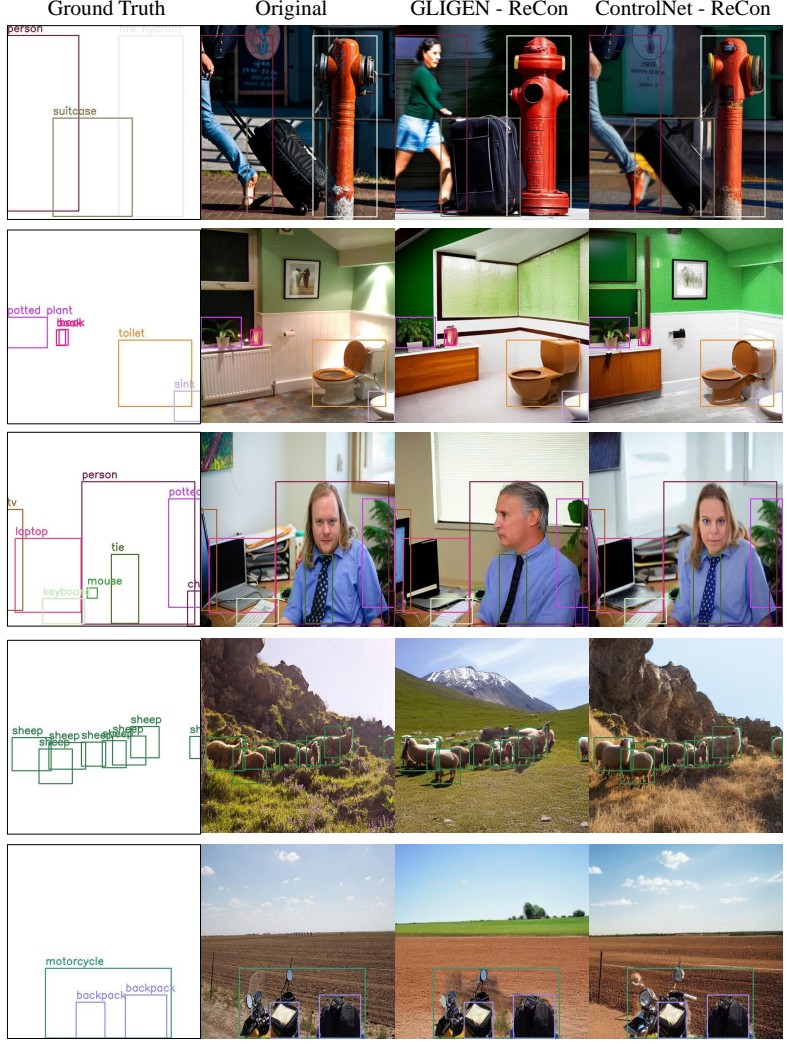

Figure 6: Visualization results of samples generated with our methods.

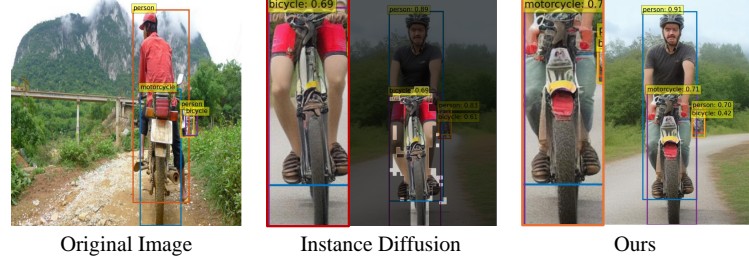

Figure 7: Successful case analysis. Existing state-of-the-art methods suffer from semantic leakage, while our approach effectively mitigates this issue.

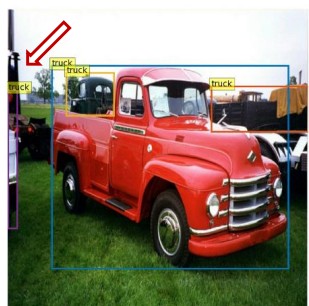 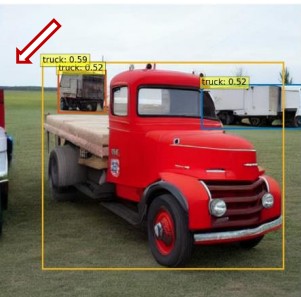

Original Image                    Ours

Figure 8: Corner case analysis. For corner cases with severe occlusion and unrealistic aspect ratios (truck on the left side of image), our method can augment them to generate new corner-case examples.

Table 8: Ablation study of region rectification at different diffusion timesteps.

| Method | mAP | $AP_{50}$ | $AP_{75}$ |
|---|---|---|---|
| ControlNet | 34.9 | 55.5 | 37.7 |
| $0.75T$ | 35.2 | 55.7 | 38.0 |
| $0.5T$ | 35.3 | 56.0 | 38.0 |
| $0.25T$ | 35.3 | 55.8 | 38.1 |
| $0.1T$ | 35.2 | 55.8 | 38.0 |
| $[0.5, 0.25]T$ | 35.3 | 55.8 | 38.2 |
| $[0.5, 0.25, 0.1]T$ | 35.4 | 56.0 | 38.4 |
| $[0.75, 0.5, 0.25, 0.1]T$ | **35.5** | **56.2** | 38.4 |
| $[0.75, 0.625, 0.5, 0.375, 0.25, 0.1]T$ | 35.5 | 56.2 | **38.5** |

of image-editing methods which modify specific regions of the original image while preserving the remainder. We compare our method against an image editing method: SDEdit (Meng et al., 2021). Notably, SDEdit applies a uniform editing strength across the entire image, which may result in certain regions being either over-enhanced or under-enhanced. As demonstrated in Table 9, our method, which leverages region-controllable data augmentation guided by a perceptual model, achieves superior performance.

Table 9: Comparison with image editing method.

| Method | mAP | $AP_{50}$ | $AP_{75}$ |
|---|---|---|---|
| Real only | 34.5 | 55.5 | 37.1 |
| SDEdit | 35.2 | 55.9 | 38.1 |
| **ReCon** | **35.5** | **56.2** | **38.4** |

### E.3 Comparison with More Detectors

As shown in Table 10, we compare additional object detectors, and the results demonstrate that our method consistently improves the performance of each.

### E.4 More Data Scaling

In the main text, we conducted data-scaling experiments under data-scarce settings. We further validated this phenomenon in few-shot scenarios. As shown in Table 11, we compare ReCon with simple duplication-based augmentation of original images under different expansion ratios. Our findings indicate that as the scale of data expansion increases, the corresponding performance improvement becomes more evident. Moreover, we observe that augmenting with duplicated original

Table 10: Trainability comparison for more detectors on COCO.

| Detector | Method | mAP | AP$_{50}$ | AP$_{75}$ |
|---|---|---|---|---|
| RetinaNet - Swin-T (Liu et al., 2021) | Real only | 34.0 | 54.0 | 35.8 |
| | **ReCon** | **35.1** | **54.9** | **37.0** |
| FCOS - R50 (Tian et al., 2019) | Real only | 36.6 | 56.0 | 39.1 |
| | **ReCon** | **37.3** | **56.4** | **39.7** |
| ATSS - R50 (Zhang et al., 2020) | Real only | 39.4 | 57.5 | 42.6 |
| | **ReCon** | **40.0** | **58.3** | **43.2** |

images provides only marginal gains, whereas our method achieves comparable results to $10\times$ duplication by using only $2\times$ generated data.

Table 11: Few-shot data augmentation results across different data scales.

| Method | mAP | AP$_{50}$ | AP$_{75}$ | AP$^m$ | AP$^l$ |
|---|---|---|---|---|---|
| Real only | 5.4 | 10.3 | 5.0 | 5.2 | 8.9 |
| *Expanded 2$\times$* | | | | | |
| Real Expansion | 5.7 | 10.9 | 5.5 | 5.8 | 9.7 |
| **ReCon** | **6.7** | **12.3** | **6.5** | **6.7** | **11.2** |
| *Expanded 5$\times$* | | | | | |
| Real Expansion | 6.1 | 11.3 | 6.1 | 6.3 | 10.2 |
| **ReCon** | **7.7** | **14.1** | **7.6** | **7.7** | **12.5** |
| *Expanded 10$\times$* | | | | | |
| Real Expansion | 6.4 | 11.5 | 6.5 | 6.3 | 10.2 |
| **ReCon** | **8.0** | **14.4** | **7.9** | **7.7** | **13.2** |

## E.5 Robustness Evaluation

We perform three independent runs with different random seeds and report the mean and standard deviation in Table 12. The results show that our method consistently outperforms the baseline and demonstrates lower variance, indicating improved stability and robustness.

Table 12: Performance stability across three runs with different random seeds. We report the mean and standard deviation of mAP and AP$_{50}$.

| Method | mAP (%) | AP$_{50}$ (%) |
|---|---|---|
| Baseline | $34.9 \pm 0.08$ | $55.3 \pm 0.12$ |
| ReCon | $\mathbf{35.5 \pm 0.05}$ | $\mathbf{56.3 \pm 0.14}$ |

### E.6 Runtime and Inference Efficiency

Table 13 reports per-sample inference times on the COCO dataset for baseline control methods and for those methods augmented with our method. All runtime measurements were collected on a single NVIDIA RTX 3090 GPU. On average, our approach introduces only an additional 0.79–1.04 seconds per sample compared to the corresponding baseline control model. Compared to the training-free layout-to-image control method LayoutGuidance, our method does not rely on backward guidance and therefore avoids the substantial time cost incurred by its optimization-based procedure. In contrast to training-based alternatives that require additional fine-tuning on downstream datasets, our framework can be applied directly at inference time without introducing extra training overhead. Finally, our method is fully compatible with structural controllable models such as ControlNet, yielding improved performance while incurring less than one second of additional inference time per sample.

We note that further acceleration is possible: replacing the grounding component with a more efficient model (e.g., EfficientSAM (Xiong et al., 2024)) or integrating memory- and compute-efficient attention libraries (e.g., xFormers (Lefaudeux et al., 2022)) should reduce the added overhead. We will include these quantitative runtime results and the above discussion in the revised manuscript to clarify the practical efficiency of our approach.

Table 13: Inference time comparison (seconds per sample) on COCO measured on an NVIDIA RTX 3090. Values in parentheses indicate the baseline time plus the additional overhead introduced by our method.

| Method | Inference time (s) |
|---|---|
| ControlNet | 2.55 |
| Layout Guidance | 12.58 |
| Instance Diffusion | 8.98 |
| ControlNet + ReCon | 3.34 (2.55 + 0.79) |
| Instance Diffusion + ReCon | 10.02 (8.98 + 1.04) |

### E.7 More Ablation Experiments

To evaluate the contribution of region-aligned cross-attention (RACA) independently from the full rectification pipeline, we augmented the InstanceDiffusion baseline with RACA while keeping all other components unchanged. The RACA layer replaces the standard cross-attention with a region-aligned cross-attention mechanism and introduces *no additional trainable parameters*, allowing direct deployment on pretrained generators without fine-tuning. Table 14 summarizes the results: adding RACA yields consistent improvements over the InstanceDiffusion baseline, confirming that region-aligned attention alone contributes meaningfully to region alignment and downstream detection performance.

Table 14: Effect of region-aligned cross-attention (RACA) within InstanceDiffusion.

| Method | mAP | $AP_{50}$ | $AP_{75}$ |
|---|---|---|---|
| InstanceDiffusion | 35.0 | 55.4 | 37.6 |
| InstanceDiffusion w/ RACA | 35.2 | 55.7 | 38.0 |

### E.8 Rationale Analysis of Our Method

Prior work has applied perception models to filter and then re-generate low-quality synthetic samples in two-step generate-then-filter pipelines (Fang et al., 2024). While effective in improving sample fidelity, such repeated generate-and-filter cycles incur substantial computational overhead. In contrast, our approach integrates grounding feedback directly into the diffusion trajectory, enabling on-the-fly rectification of misaligned regions within a single forward pass. This integrated rectification avoids repeated re-sampling and produces high-quality, aligned image-label pairs far more efficiently than multi-round generate-and-filter procedures.

Our rectification mechanism (RGR) identifies mismatched regions via grounding-based IoU matching between generated regions and target annotations, and selectively injects controlled perturbations into those mismatched regions during the diffusion trajectory. This targeted intervention encourages subsequent denoising steps to correct region appearance and spatial extent while leaving well-aligned regions largely untouched.

The proposed design achieves three complementary properties that are critical for large-scale synthetic data pipelines:

- **Usability.** The method does not require model retraining, multi-round generation, or post-hoc filtering; it operates during a single forward pass and can be applied to pretrained generators out-of-the-box.
- **Effectiveness.** Experiments across multiple datasets and generator architectures show consistent improvements in both image quality and detection metrics.
- **Efficiency.** The rectification strategy introduces limited additional computational overhead to inference, making it suitable for large-scale synthetic data generation.

These benefits stem from two lightweight, jointly designed modules (region-aligned attention and grounding-guided rectification) that together ensure improved alignment without compromising generation speed or flexibility. To reach the final design we explored and quantitatively compared a variety of correction and alignment strategies; the selected combination offers a favorable trade-off between alignment quality, computational cost, and ease of integration.

In summary, the combination of on-the-fly grounding feedback, parameter-free region-aligned attention, and targeted rectification yields a practical and effective solution for producing well-aligned image-label pairs. The plug-and-play compatibility with pretrained generators and consistent empirical gains across baselines and datasets underscore the substantive contributions of this work.

### E.9 Further Analysis of Rectified Regions

We analyzed the relationship between object area and the likelihood of rectification under our proposed Region-Guided Rectification strategy. Specifically, we examined 500 objects identified as requiring rectification and plotted their area distribution using kernel density estimation, as shown in Figure 9. The results reveal that smaller regions are more likely to be rectified. This is because small objects are generally more difficult to synthesize accurately, diffusion models tend to generate artifacts or errors in such regions.

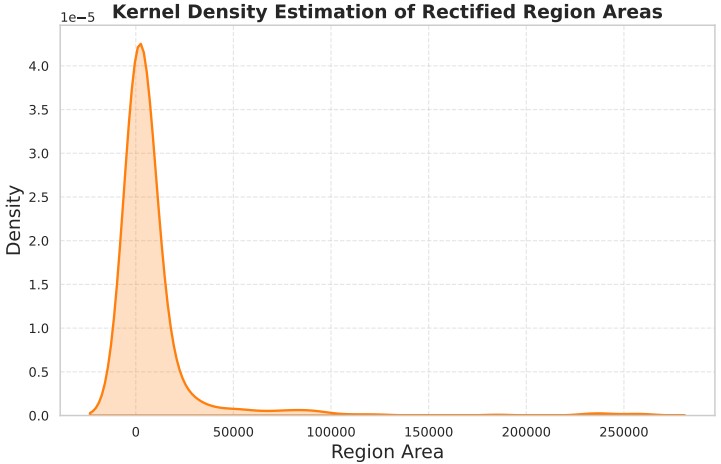

Figure 9: Kernel density estimation of rectified region areas.

