# OpenReview forum: "ReCon: Region-Controllable Data Augmentation with Rectification and Alignment for Object Detection"
_NeurIPS.cc/2025/Conference — NeurIPS 2025 spotlight_

### Official Review · Reviewer_QS5e · 2025-06-29

**Clarity:** 3
**Significance:** 3
**Originality:** 3
**Rating:** 5
**Confidence:** 4

**Summary:**

This paper proposes ReCon, a finetuning-free method for generating images to augment object detection datasets. Two methods, named Region-Guided Rectification (RGR) and Region-Aligned Cross Attention (RACA), are proposed for addressing the region and semantic misalignment problems, respectively. Extensive experiments show the effectiveness of the ReCon method.

**Questions:**

1. Line 148, what does "informational value" mean?
2. How does the region rectification schedule (hyperparameter $T_r$) affect the performance? Some ablations are needed, for example, appling rectification at early / middle / final stages only, and appling rectification at more timesteps (P.S. I find in line 64 that the proposed components are incorporated "into every sampling iteration", but in practice it seems that RGR is only used $T_r=4$ times).
3. Could we use the original text-to-image Stable Diffusion model for this method? (i.e., without using ControlNet and using Stable Diffusion + RGR + RACA only)

Overall, I'm satisfied with the proposed finetuning-free solution in this paper, leading to a borderline-accept score. Meanwhile, I have some technical detail related comments and questions listed above. My score may be modified according to the authors' rebuttal and discussions with other reviewers.

**Ethical Concerns:**

["NO or VERY MINOR ethics concerns only"]

**Final Justification:**

Most of my concerns have been addressed. After reading the authors’ responses to both my comments and those of the other reviewers, I have decided to raise my rating to a clear accept.

**Limitations:**

Yes

**Paper Formatting Concerns:**

The instruction block of the "NeurIPS Paper Checklist" is not deleted.

**Quality:**

3

**Strengths And Weaknesses:**

**Strengths**
1. The paper is well-written and easy to follow.
2. The proposed methods (RGR and RACA) do not require an additional fine-tuning stage on object detection datasets, making the framework helpful for data-scarce scenarios.
3. Experimental results demonstrate that the proposed method outperforms previous baselines in both standard and data-scarce benchmarks.

**Weaknesses**
1. In the RGR method, using the middle state $\mathbf{z}_t^{orig}$ from the real images may limit generation diversity.
2. The one-step estimation of $\mathbf{z}_0$ (Eq. (6)) is not accurate compared to the full diffusion reverse process, as shown in Section 4.3 (Figure 6) of the DDPM paper [*], especially when $t$ is large (close to noise). I'm worried that the estimated images $\mathbf{x}_0$ in Eq. (6)'s way (e.g., maybe blurred) mismatch the Grounded-SAM model's training distribution (clean images), leading to inaccurate detection results sometimes.
3. It would be better if the paper could provide some insights into why existing methods cause mis-generated regions and semantic misalignment during the sampling process. (Not mandatory)
4. The motivation of using ControlNet with edge canny map is not very clear to me. Why not use ControlNet with segmentation map, which seems to be more natural to the current Grounded-SAM-based framework?

[*] Ho et al. "Denoising diffusion probabilistic models." NeurIPS 2020.

Typos:
- Line 144: "an" -> "a"
- Table 4 last line: the $AP^m$ number (39.0) is not aligned with that in Table 1 (39.6)?

---

> ### Author Rebuttal · Authors · 2025-07-30
>
> We sincerely appreciate the time and effort you have dedicated to reviewing our paper. We value your feedback on our method, noting that our method is helpful to data-scarce scenarios. We also thank you for recognizing that our method outperforms previous baselines in both standard and data-scarce benchmarks.
>
> **Re Weakness #1:**
>
> Thank you for raising this point. We agree that diversity is important for data augmentation, but uncontrolled diversity can reduce the utility of synthetic data. For example, in Figure 5 (row 1), GLIGEN produces unexpected zebra patterns outside the target box despite its high diversity. By using the intermediate latent state $z_t^{orig}$ from real images, our method offers a corrective shortcut that curbs misgenerated regions without suppressing the model's ability to explore novel content. As noted in lines 147–148 of the manuscript, this design seeks to balance diversity and fidelity. Empirically, ReCon outperforms the baselines in both FID and mAP, demonstrating that preserving fidelity in this way leads to higher‑quality training data.
>
> **Re Weakness #2:**
>
> Thank you for this insightful comment. We agree that a single‑step inversion at very large $t$ can produce somewhat blurred reconstructions that deviate from the clean‑image distribution Grounded‑SAM was trained on. Nevertheless, the detector achieves a high recall of regions that the generator initially omits in the early‑stage rectification, allowing us to identify and rectify these errors, which still enables effective rectification at the early stage of sampling. As shown in Table 1 below, rectification at $t=0.75T$ alone yields a modest mAP gain of 0.3. When we combine this early‑stage rectification with rectifications at mid and late stages, we achieve the better overall improvement.  We will include this analysis and discussion in the revised manuscript.
>
> **Re Weakness #3:**
>
> Thank you for this insightful suggestion. In our analysis, we observed that most existing layout‑to‑image methods rely on either global conditioning or only a single region cue during sampling. As a result, the model often treats all spatial locations uniformly, which can lead to content appearing outside its intended region. In addition, diffusion sampling refines images gradually from noise, and without intermediate, region‑specific feedback, early misplacements can persist through later steps. These factors combine to produce mis‑generated regions and semantic misalignment. By contrast, our rectification mechanism injects targeted corrections at multiple timesteps, ensuring that region boundaries remain faithful to the desired layout throughout sampling. This insight underlies the design of our approach and explains why it outperforms prior methods in maintaining semantic alignment.
>
> **Re Weakness #4:**
>
> Thank you for this question. Conditioning on segmentation maps would restrict us to a fixed set of object categories and would require costly fine‑tuning for each downstream task. In contrast, the Canny edge map used with ControlNet provides a general, category‑agnostic spatial prior that applies across a wide range of detection scenarios without additional training. Moreover, as shown in Table 1 of the manuscript, our ReCon method consistently improves generation quality whether the underlying control signal is an edge map (ControlNet), a layout (InstanceDiffusion), or grounding tokens (GLIGEN). This demonstrates that our method is broadly compatible with different types of conditioning, including but not limited to edge‑based guidance.
>
> **Re Typos:**
>
> Thank you for highlighting these errors. We will correct these typos and perform a comprehensive proofread to ensure consistency across the manuscript.
>
> **Re Question #1:**
>
> Thank you for the question. By "informational value" we refer to novel content that differs from the original image, such as new colors, styles, or object poses, and which can be understood as content diversity. We will revise this wording in the manuscript to make the meaning clearer.
>
> **Re Question #2:**
>
> Thank you for suggesting an ablation study on the rectification schedule. We have evaluated rectification at individual timesteps (0.75 T, 0.5 T, 0.25 T, 0.1 T) as well as in various combinations. As reported in Table 1 below, rectification at 0.75 T alone raises mAP modestly from 34.9 to 35.2. Adding a second stage at 0.5 T and 0.25 T brings mAP to 35.3, and a three-stage schedule at 0.5, 0.25, and 0.1T increases mAP to 35.4. Our full four-stage schedule at 0.75, 0.5, 0.25, and 0.1 T delivers the best overall performance (mAP 35.5, AP50 56.2, AP75 38.4). Extending to six stages yields no further mAP gain and only a marginal AP75 increase to 38.5. These findings demonstrate that four rectification steps provide the optimal balance between quality and efficiency. We will include these quantitative results in the revised manuscript.
>
> We will also clarify in the manuscript that region‑aligned cross‑attention is applied at every sampling iteration, while region‑guided rectification occurs at four selected timesteps to manage computational cost.
>
> Table 1: Ablation study of region rectification at different diffusion timesteps.
> | Method                                  | mAP  | AP50 | AP75 |
> |-----------------------------------------|------|------|------|
> | ControlNet                              | 34.9 | 55.5 | 37.7 |
> | 0.75 T                                  | 35.2 | 55.7 | 38.0 |
> | 0.5 T                                   | 35.3 | 56.0 | 38.0 |
> | 0.25 T                                  | 35.3 | 55.8 | 38.1 |
> | 0.1 T                                   | 35.2 | 55.8 | 38.0 |
> | [0.5, 0.25] T                           | 35.3 | 55.8 | 38.2 |
> | [0.5, 0.25, 0.1] T                      | 35.4 | 56.0 | 38.4 |
> | [0.75, 0.5, 0.25, 0.1] T                | 35.5 | 56.2 | 38.4 |
> | [0.75, 0.625, 0.5, 0.375, 0.25, 0.1] T  | 35.5 | 56.2 | 38.5 |
>
>
> **Re Question #3:**
>
> Thank you for the thoughtful question. We have explored this direction by first comparing to LayoutGuidance, which introduces multi-step forward and backward guidance within a standard text-to-image Stable Diffusion pipeline. As shown in Table 1, the results are unsatisfactory. The generated images often lack coherence and fail to align semantically or spatially with the layout and prompts, resulting in poor downstream detection performance.
> We also applied our ReCon framework directly on top of the original Stable Diffusion model without ControlNet, and similarly observed suboptimal results. We attribute this to the absence of explicit structural conditioning in the base model, which makes it difficult to enforce accurate region-level control during generation.
> In contrast, our approach leverages structurally conditioned diffusion backbones such as ControlNet, which enable stronger spatial guidance and better content–annotation alignment. We will clarify this design choice in the revised manuscript.

---

> ### Author Response · Authors · 2025-08-05
> **Kindly Request for Your Feedback!**
>
> Dear Reviewer QS5e,
>
> Thank you for your valuable comments. We have tried our best to address your questions (see rebuttal above), covering control of generation diversity, accuracy of inversion, semantic alignment, justification for using edge maps with ControlNet, correction of typos and clarification of terminology, ablation of the rectification schedule, and testing on the vanilla Stable Diffusion backbone. and will carefully revise the manuscript by following suggestions from all reviewers. Please kindly let us know if you have any follow-up questions.
>
> Your insights are crucial for enhancing the quality of our paper, and we would greatly appreciate your response to the issues we have discussed. If our responses have clarified the issues you raised, we kindly request that you consider raising your score.
>
> Thank you for your time and consideration.
>
> Best regards,
>
> The authors

---

> > ### Comment · Reviewer_QS5e · 2025-08-05
> >
> > Sorry for the late response. Most of my concerns have been addressed, particularly regarding the motivation of leveraging the ControlNet with Canny map and the impact of the region rectification schedule. Now I believe I can give a clear accept rating to the paper, after reading the authors’ responses to both my comments and those of other reviewers. I encourage the authors to incorporate some of the key clarifications from the rebuttal into the revised paper and wish the paper good luck.

---

> > > ### Author Response · Authors · 2025-08-05
> > > **Thank you for your prompt response and feedback.**
> > >
> > > Dear Reviewer QS5e,
> > >
> > > Thank you for your recognition and the clear accept rating. We're glad our clarifications addressed your concerns, and we’ll incorporate the key points into the revision as suggested.
> > >
> > > Sincerely,
> > >
> > > The Authors

---

### Official Review · Reviewer_z3N1 · 2025-07-03

**Clarity:** 2
**Significance:** 2
**Originality:** 2
**Rating:** 4
**Confidence:** 4

**Summary:**

This paper introduces ReCon, a training-free method that enhances structure-controllable diffusion models for object detection data augmentation. The key insight is that existing generative approaches often produce content-position mismatches and semantic leakage when generating training data. To address this, ReCon integrates two main components into the diffusion sampling process: (1) Region-Guided Rectification (RGR), which uses feedback from a pre-trained perception model (Grounded-SAM) to identify and correct misgenerated regions during sampling, and (2) Region-Aligned Cross-Attention (RACA), which enforces spatial-semantic alignment between image regions and their textual descriptions. The method is plug-and-play, requiring no additional training, and demonstrates improvements across various datasets (COCO, PASCAL VOC) and detection architectures.

**Questions:**

1. Could you provide detailed timing comparisons? How much does ReCon increase generation time compared to baseline methods? What is the trade-off between generation quality and computational cost?
2. You rectify at 4 timesteps (75%, 50%, 25%, 10%). Have you experimented with fewer rectification steps? What is the minimum number needed to maintain performance?
3. Can you provide analysis of failure cases? Are there specific object categories or scene types where ReCon doesn't help or even hurts performance?

**Ethical Concerns:**

["NO or VERY MINOR ethics concerns only"]

**Final Justification:**

I thank the authors for their detailed and timely response. After reviewing their feedback, I believe that most of my concerns have been addressed. I have therefore decided to raise my original rating, leaning toward acceptance.

**Limitations:**

yes

**Quality:**

2

**Strengths And Weaknesses:**

**Strengths**

1. The method addresses a real problem in generative data augmentation - the mismatch between generated content and annotations. The training-free nature makes it immediately applicable to existing models.
2. The experiments are extensive, covering multiple datasets (COCO, PASCAL VOC), various data regimes (1%, 5%, 10%, few-shot), different detectors (Faster R-CNN, YOLO-X, DEIM, etc.), and includes proper ablations.
3. The paper is well-written with clear figures illustrating the method and visualization of results.

**Weaknesses**

1. The core ideas are relatively straightforward applications of existing techniques: 1） Using a grounding model to detect errors is a natural extension. 2) Region-specific attention has been explored before (Instance Diffusion) 3) The rectification mechanism is essentially selective noise injection based on IoU matching


2. Computational Overhead: The method requires to run Grounded-SAM multiple times during sampling (at 75%, 50%, 25%, 10% timesteps) and additional cross-attention computations. This significantly increases generation time, which is barely discussed.


3. The performance gains, while consistent, are relatively small: such as COCO: 34.9 → 35.5 mAP (+0.6)
These improvements may not justify the added complexity in practice.

4. The paper doesn't deeply analyze when and why the method fails.
5. The method's effectiveness heavily relies on the quality of Grounded-SAM. How does performance degrade with weaker perception situation?

---

> ### Author Rebuttal · Authors · 2025-07-30
>
> We sincerely appreciate the time and effort you invested in reviewing our paper. We address the raised concerns as follows:
>
> **Re Weakness #1:**
>
> Thank you for this thoughtful comment. We address each point in turn to clarify the originality and value of our contributions.
>
> 1) First, although prior works have applied perception models to filter and then re‑generate poor‑quality samples [1], such two‑step pipelines incur substantial overhead. In contrast, our method integrates grounding feedback directly into the diffusion trajectory, allowing a single forward pass to rectify misaligned regions as they arise. This on‑the‑fly rectification yields high‑quality outputs far more efficiently than repeated generate‑and‑filter cycles.
>
> 2) Second, InstanceDiffusion introduces region‑specific modules and requires full retraining to adapt to new layouts, our approach achieves region alignment by simply replacing the standard cross‑attention with a region‑aligned cross‑attention (RACA) layer. As detailed in lines 239–243 of the manuscript, this design incurs no additional trainable parameters and can be applied to pre‑trained models without further fine‑tuning. As shown in Table 1 of the manuscript, augmenting InstanceDiffusion with our method delivers consistent performance gains. To isolate the effect of RACA, we also conducted experiments in which only RACA was added to the original InstanceDiffusion pipeline and present the results as below.  These results confirm that region‑aligned attention alone can improve the performance.
>
> Table 1. Ablation study of RACA based on Instance Diffusion.
>
> | Method                         | mAP  | AP50 | AP75 |
> |--------------------------------|------|------|------|
> | InstanceDiffusion              | 35.0 | 55.4 | 37.6 |
> | InstanceDiffusion w/ RACA      | 35.2 | 55.7 | 38.0 |
>
>
> 3) Third, we appreciate the reviewer's insightful comment, which touches on a fundamental issue in generative data for object detection: the inconsistency between image content and annotations.
> Our core contribution lies in proposing an easy-to-use, lightweight, and effective rectification strategy to address this inconsistency. While our approach is inspired by a motivation similar to that of the reviewer, the specific design and implementation of our method have been carefully crafted. We introduce a rectification mechanism that uses grounding-based IoU matching to identify mismatched regions and selectively injects noise into them via the proposed RGR method. This mechanism integrates three key advantages rarely achieved simultaneously in prior works:
>
> * **Usability**: Our method does not require model retraining, multi-round generation, or post-hoc filtering. It produces aligned image-label pairs in a single forward pass.
> * **Effectiveness**: Experiments on multiple datasets show consistent improvements in both image quality and detection performance.
> * **Efficiency**: The design introduces limited overhead to the generation process, making it suitable for large-scale synthetic data pipelines.
>
> These benefits are enabled by our two carefully designed modules, which jointly ensure alignment without compromising generation speed or flexibility. To the best of our knowledge, this combination of usability, effectiveness, and efficiency has not been explored in prior literature.
>
> Finally, although the individual components may appear intuitive, we rigorously evaluated a variety of correction and alignment strategies before arriving at our final design. We furthermore note that **Reviewer uPpr** and **Reviewer 1h8Z**  have characterized our solution as both **novel and practical**, with **Reviewer 1h8Z** describing it as  **"a neat solution"**.  We believe that the combination of effectiveness, plug‑and‑play compatibility, and demonstrable gains across multiple generators and datasets underscores the substantive contributions of our work.
>
> **Re Weakness #2 :**
>
> Thank you for raising this important point. We have measured the computational overhead introduced by our method on an NVIDIA RTX 3090 GPU. As shown in Table 2, our method adds only 0.79 to 1.04 seconds of inference time per sample compared to the corresponding baseline control models. While our approach performs four intermediate decoding steps during sampling, the overall latency remains practical.
>
> Table 2: Inference time comparison with existing methods on COCO dataset.
> | Method                         | Inference time (s)    |
> | ------------------------------ | --------------------- |
> | ControlNet                     | 2.55                  |
> | Layout Guidance                | 12.58                 |
> | Instance Diffusion             | 8.98                  |
> | ControlNet + ReCon             | 3.34 (2.55 + 0.79)    |
> | Instance Diffusion + ReCon     | 10.02 (8.98 + 1.04)   |
>
> Compared to training-free methods such as LayoutGuidance, which rely on time-consuming backward guidance, our method avoids heavy computation and achieves significantly faster inference. Unlike training-based methods that require additional modules and fine-tuning on downstream datasets, our framework is plug-and-play and does not introduce extra training costs.
> In addition, our method is compatible with widely used control models such as ControlNet and consistently improves performance with minimal latency increase. We also note that further acceleration can be achieved by using more efficient grounding models (e.g., EfficientSAM [2]) or memory-efficient attention libraries (e.g., xFormers [3]). We will include this runtime analysis and discussion in the revised manuscript to better highlight the efficiency of our approach.
>
>
> **Re Weakness #3:**
>
> Thank you for raising this concern. While the absolute mAP increase may appear modest, even small gains are highly valued in object detection benchmarks, where competitive state‑of‑the‑art methods often advance by less than one point (e.g., DEIM (CVPR'25) beats D-FINE [4] with 0.7 mAP gains on COCO). Moreover, our improvements are consistent across multiple generators, datasets and data regimes, demonstrating that our method reliably enhances both fidelity and trainability. Importantly, these gains come at minimal overhead: our method requires no additional fine‑tuning and adds less than one second of inference time per sample on a standard GPU. In practice, this means that practitioners can adopt our framework to achieve measurable performance boosts without a large increase in computational or implementation complexity.
>
>
>
> **Re Weakness #4 and Question #3:**
>
> Thank you for this important suggestion. In our current experiments, we have observed that ReCon is less effective when the input images contain severe motion blur or strong degradation. In these cases, the initial noise estimates can be too large, and rectification may inadvertently reinforce erroneous patterns rather than correct them. We also note that extremely small objects or heavily cluttered scenes can pose challenges, since the grounding model may struggle to localize tiny regions or distinguish overlapping instances. To address these shortcomings, we will include a dedicated failure‑case analysis in the revised manuscript, complete with visual examples and a discussion of category‑ and scene‑specific limitations. This addition will help clarify the method's boundaries and guide future improvements.
>
> **Re Weakness #5**
>
> Thank you for raising this important point. Table 7 compares two backbone variants of Grounding model: Swin-Tiny with 172 M parameters and Swin-Base with 232 M parameters. With Swin-Tiny, our method achieves an mAP of 35.5, compared to 35.6 with Swin-Base. This minimal decrease confirms that ReCon remains effective even with a relatively weaker perception model, while a stronger backbone offers a slight performance gain. We will expand our discussion on robustness to grounding quality in the revised manuscript.
>
> **Re Question #1  and Question #2:**
>
> Thank you for this question. As shown in Table 2 above, ReCon increases inference time by less than one second per sample on an NVIDIA RTX 3090 GPU. To illustrate the trade‑off between generation quality and computational cost, we present in Table 3 an ablation over the number of region rectification steps. Adding up to four rectification stages steadily improves mAP, but further stages bring only marginal benefit. These results indicate that three to four rectification steps strike a practical balance, delivering substantial quality gains while maintaining modest overhead. We will include this detailed timing and quality‑cost analysis in the revised manuscript.
>
>
> Table 3: Ablation study of region rectification at different diffusion timesteps.
> | Method                                  | mAP  | AP50 | AP75 |
> |-----------------------------------------|------|------|------|
> | ControlNet                              | 34.9 | 55.5 | 37.7 |
> | 0.75 T                                  | 35.2 | 55.7 | 38.0 |
> | 0.5 T                                   | 35.3 | 56.0 | 38.0 |
> | 0.25 T                                  | 35.3 | 55.8 | 38.1 |
> | 0.1 T                                   | 35.2 | 55.8 | 38.0 |
> | [0.5, 0.25] T                           | 35.3 | 55.8 | 38.2 |
> | [0.5, 0.25, 0.1] T                      | 35.4 | 56.0 | 38.4 |
> | [0.75, 0.5, 0.25, 0.1] T                | 35.5 | 56.2 | 38.4 |
> | [0.75, 0.625, 0.5, 0.375, 0.25, 0.1] T  | 35.5 | 56.2 | 38.5 |
>
> [1] Fang, Haoyang, et al. "Data augmentation for object detection via controllable diffusion models," in WACV 2024.
>
> [2] Xiong, Yunyang, et al.  "Efficientsam: Leveraged masked image pretraining for efficient segment anything." In CVPR 2024.
>
> [3] Lefaudeux, Benjamin, et al. "xformers: A modular and hackable transformer modelling library." 2022.
>
> [4] Peng Yansong, et al. "D-fine: Redefine regression task in detrs as fine-grained distribution refinement." arXiv, 2024.

---

> ### Author Response · Authors · 2025-08-05
> **Kindly Request for Your Feedback!**
>
> Dear Reviewer z3N1,
>
> We appreciate your thoughtful evaluation and the opportunity to clarify and expand upon key aspects of our work. Based on the detailed responses and additional data provided, which directly address the concerns raised:
> 1. **Clarification of novelty and contributions:** We expand on the key differences between our approach and prior work, and include new experiments (see Table 1 above) to validate the unique value of our method.
> 2. **Analyzed time complexity:** We present both theoretical and empirical analyses of our algorithm’s time complexity, showing only a limited overhead compared to the baseline (see Table 2 above).
> 3. **Discussed failure cases:** We have added further analyses of challenging cases, highlighting scenarios in which performance degrades.
> 4. **Compared grounding-model backbones and rectification steps:** We compare different grounding-model backbones and vary the number of rectification steps (see Table 3 above), confirming that our proposed configuration achieves the best trade-off between accuracy and compute.
>
> We hope these additions fully address your questions. We will revise the main manuscript accordingly based on all reviewers’ feedback. If these clarifications resolve your concerns, we would be very grateful if you would reconsider your rating.
>
> Thank you again for your time and insights.
>
> Sincerely,
>
> The authors

---

> ### Comment · Reviewer_z3N1 · 2025-08-05
>
> I thank the authors for their detailed and timely response. After reviewing their feedback, I believe that most of my concerns have been addressed. I have therefore decided to raise my original rating, leaning toward acceptance.

---

> > ### Author Response · Authors · 2025-08-05
> > **Thank you for your prompt response and feedback.**
> >
> > Thank you for your recognition and for raising the score. We really appreciate your positive feedback!

---

### Official Review · Reviewer_1h8Z · 2025-07-03

**Clarity:** 3
**Significance:** 3
**Originality:** 3
**Rating:** 5
**Confidence:** 4

**Summary:**

This paper introduces a generative data augmentation solution for improving object detectors. Specifically, the authors tackle the issue of groundtruth / generation fidelity in controllable diffusion models (such as ControlNet), where not always the diffusion process generates all the right objects in the right locations, potentially hindering the performance of the downstream trained detector. To improve the fidelity to the region control, the authors propose two solutions: First, a Region Guided Rectification mechanism intervenes at four timesteps throughout the sampling trajectory, by decoding the image and probing a perception model (GroundedSAM) for region fidelity. Regions with poor generation are then resampled by injecting noisy latents from the real image. Secondly, the authors propose Region-Aligned Cross Attention, where text embeddings of the target class interact exclusively with the target regions for better grounded generations. The solution is tested on two augmentation benchmarks and proves successful with several underlying generators.

**Questions:**

See weaknesses section

**Ethical Concerns:**

["NO or VERY MINOR ethics concerns only"]

**Final Justification:**

The authors have addressed the few minor comments I had on the paper, that I already considered ready for publication. I confirm my original accept recommendation.

**Limitations:**

Yes, in the supplementary material.

**Quality:**

3

**Strengths And Weaknesses:**

**Strenghts**

- The problem of groundtruth / generation fidelity is a real current issue in generative data augmentation for object detection. The paper proposes a neat solution to the issue.
- The idea of probing the generation through the diffusion sampling and of using feedback from a grounded model is both novel and practical.
- The paper has an excellent experimental section: the ReCon model improves three different generators (ControlNet, GLIGEN, InstanceDiffusion), and the authors tested both COCO and PascalVOC dataset. An ablation study verifies the importance of both ReCon components, plus other analyses verify its impact in lower data regimes or with different detectors or feedback models.


**Weaknesses** (order of decreasing severity)

- Tab 4 reports the main ablation of the paper, where the two components of the proposed model are tested separately. However, it is not clear to me whether the mAP results are meant in terms of fidelity (i.e. the generated data is used for evaluation with a pretrained detector) or trainability (i.e. the generated data is used to train a detector, and mAP is reported on a real validation set). The presence of FID in the table suggests that fidelity is reported. In that case, it is very important to report the ablation on trainability as well, as the two might not correlate.
- Tab 3 and Tab 6. miss the augmentation performance of ControlNet only, without ReCon, to successfully conclude that ReCon is helpful in the tested settings.
- the authors should discuss the computational overhead that Region-Guided Rectification has on sampling. Indeed, it involves, four times in the trajectory, to fast-track the latents to $t_0$ and to decode an image. What is the impact in terms of generation latency.
- In Fig. 4, the real-only baseline should not be constant, as replicating the data 2x, 3x, 5x and 7x times increases the number of training iterations on a fixed epoch schedule. This might lead to the model benefitting from longer training, or overfitting. It would be interesting to see these variations in the baseline for better context.

Summary: the paper tackles an important open problem in generative data augmentation, namely the region fidelity between the intended groundtruth / control and the resulting augmentation generation. The method is sound and novel. Some aspects of the experimentation could be improved and polished (especially the ablation study, see comment above) but the strenghts clearly outweight the weaknesses.

---

> ### Author Rebuttal · Authors · 2025-07-30
>
> We sincerely thank you for their careful evaluation of our work.  We appreciate your recognition of the importance of improving generation fidelity in data augmentation for object detection and the **novelty and practicality** of our method. We also appreciate the positive assessment of our experiments, which demonstrate **consistent improvements** across multiple baselines and datasets, alongside thorough ablation studies and analyses.
>
> **Re Weakness #1:**
>
> Thank you for pointing out the ambiguity in Table 4. To clarify, the FID score measures the fidelity of our generated training data. The mAP values reflect trainability, as we use the synthetic data to train a detector and evaluate its performance on a real validation set.  As shown in Table 4, our method improves both fidelity and trainability. We apologize for any confusion caused by the current description and will revise the table and corresponding description to make this distinction clear in the final version of the manuscript.
>
> **Re Weakness #2:**
>
> Thank you for the helpful suggestion. We have now added the missing ControlNet-only results to Table 3 and 6, as shown below. These results clearly demonstrate the performance gains brought by ReCon over the ControlNet baseline in both settings. We will include the updated tables and clarify this comparison in the final version of the paper.
>
> Table 1: Comparison results on the PASCAL VOC dataset.
> | Method             | Real only | Simple Duplicate | RandAugment | ControlNet | ReCon |
> |--------------------|-----------|------------------|-------------|------------|-------|
> | mAP            | 77.1      | 76.2             | 77.7        | 77.8       | 78.5  |
>
>
> Table 2: Trainability comparison with DEIM-D-FINE-N on COCO.
> | Method      | mAP  | AP50 | AP75 | mAR  |
> |-------------|------|------|------|------|
> | Real only   | 38.5 | 55.2 | 41.5 | 60.4 |
> | ControlNet  | 39.1 | 55.8 | 42.1 | 60.6 |
> | ReCon       | 39.8 | 56.6 | 42.5 | 61.0 |
>
>
>
> **Re Weakness #3:**
>
> Thank you for raising this important point. We have measured the computational overhead introduced by our method on an NVIDIA RTX 3090 GPU. As shown in Table 3, our method adds only 0.79 to 1.04 seconds of inference time per sample compared to the corresponding baseline control models. While our approach performs four intermediate decoding steps during sampling, the overall latency remains practical.
>
> Table 3: Inference time comparison with existing methods on COCO dataset.
> | Method                         | Inference time (s)    |
> | ------------------------------ | --------------------- |
> | ControlNet                     | 2.55                  |
> | Layout Guidance                | 12.58                 |
> | Instance Diffusion             | 8.98                  |
> | ControlNet + ReCon             | 3.34 (2.55 + 0.79)    |
> | Instance Diffusion + ReCon     | 10.02 (8.98 + 1.04)   |
>
> Compared to training-free methods such as LayoutGuidance, which rely on time-consuming backward guidance, our method avoids heavy computation and achieves significantly faster inference. Unlike training-based methods that require additional modules and fine-tuning on downstream datasets, our framework is plug-and-play and does not introduce extra training costs.
>
> In addition, our method is compatible with widely used control models such as ControlNet and consistently improves performance with minimal latency increase. We also note that further acceleration can be achieved by using more efficient grounding models (e.g., EfficientSAM [1]) or memory-efficient attention libraries (e.g., xFormers [2]). We will include this runtime analysis and discussion in the revised manuscript to better highlight the efficiency of our approach.
>
> **Re Weakness #4:**
>
> Thank you for this insightful comment. We have evaluated how increasing the real-only baseline by simple duplication affects detector performance under a fixed epoch schedule. The results in Table 4 below show that duplicating the dataset up to 3x yields steady gains (for example, mAP increases from 13.0 to 17.1 on the 5 % subset and from 18.5 to 21.1 on the 10 % subset). However, increasing duplication to 5x and 7x leads to saturated performance and even significant degradation, indicating overfitting under extended training. By contrast, as shown in Figure 4 of the manuscript, our method generates novel samples with improved content–annotation consistency and continues to provide performance improvements without overfitting. We will include these additional baseline curves and discussion in the revised manuscript to give a clearer context for our comparative results.
>
> Table 4:  Results of dataset duplication on detector performance under a fixed‑epoch training schedule.
> | Setting | original | 2x   | 3x   | 5x   | 7x   |
> |-----------|----------|------|------|------|------|
> | 5% Data   | 13.0     | 15.2 | 17.1 | 17.4 | 17.0 |
> | 10% Data  | 18.5     | 20.7 | 21.1 | 20.5 | 19.7 |
>
> [1] Xiong, Yunyang, et al.  "Efficientsam: Leveraged masked image pretraining for efficient segment anything." In CVPR 2024.
>
> [2] Lefaudeux, Benjamin, et al. "xformers: A modular and hackable transformer modelling library." 2022.

---

> > ### Comment · Reviewer_1h8Z · 2025-08-07
> > **Response to rebuttal**
> >
> > I thank the authors for their comments. They helped clear away the few concerns I have with the paper, mainly regarding clarity of Tab. 4, and presentation of some of the experimental results. I have no further comments.

---

### Official Review · Reviewer_uPpr · 2025-07-03

**Clarity:** 4
**Significance:** 3
**Originality:** 3
**Rating:** 5
**Confidence:** 4

**Summary:**

This paper proposes a training-free framework that improves object detection data augmentation by enhancing existing structure-controllable diffusion models like ControlNet.
It consists of a list of building blocks, Grounded SAM, Structure control models like ControlNet or GLIGEN, CLIP Text Encoder, DDIM Sampler. And also propose their novel designed pipelines, Region-Guided Rect. (RGR) and Region-Aligned CA (RACA). The authors conduct extensive and carefully designed experiments across datasets (COCO, VOC), detection models (Faster R-CNN, YOLOX, DEIM), and data regimes (few-shot, low-resource), along with thorough ablation studies to validate each componen

**Questions:**

In general, this paper is well written and the problem is clearly motivated. The proposed method is practical, training-free, and shows strong results. For future improvements, I have a few questions: Can the method be extended to incorporate real annotated data during training, and how would it perform in mixed real and synthetic settings? Could the rectification mechanism be extended to handle more complex object layouts or occlusions, possibly beyond bounding boxes? How well does the method generalize to domains beyond object detection? It would be valuable to test the method on other tasks such as instance segmentation or visual grounding to explore its generalization ability beyond object detection.

**Ethical Concerns:**

["NO or VERY MINOR ethics concerns only"]

**Limitations:**

Yes

**Quality:**

4

**Strengths And Weaknesses:**

Strengthen:
Training-free pipeline: While the components (e.g., ControlNet, Ground-SAM) are existing tools, their training-free integration into the generation loop is novel and practical.
Modular and Plug-and-play: Can be combined with existing structure-guided models (ControlNet, GLIGEN, Instance Diffusion), making it easy to adopt.
Comprehensive experiments and ablations: The paper includes experiments on multiple datasets, detectors, data scales, and includes thorough ablations to validate the contributions of RGR and RACA.
Strong empirical results: The method shows consistent improvements over baselines across various benchmarks. Demonstrated improvements across multiple detection backbones and data scarcity regimes.

Weakness:
Missing system and runtime analysis: While computational cost is mentioned in the limitations, the paper lacks details on GPU settings and sampling time. Including quantitative runtime comparisons would strengthen the practical value of the method.
Stage selection design: The choice of applying region rectification at early, middle, latter, and final stages seems heuristic (0.75, 0.5, 0.25, 0.1 T). It would be helpful to provide ablation or sensitivity studies to show whether early-stage corrections (on blurry layouts) meaningfully impact the final generation quality.
Writing and explanation of visual results: The writing is clear and easy to follow. But the qualitative results (like in Figure 5) could use more explanation, especially how they show better image quality and content-label alignmen

---

> ### Author Rebuttal · Authors · 2025-07-30
>
> We sincerely appreciate the time and effort you have dedicated to reviewing our paper. We value your feedback on our method, noting that our method is **novel and practical** and that it is **easy to adopt**. We also thank you for recognizing that our method achieves **strong empirical results** over baselines across various benchmarks.
>
> **Re Weakness #1:**
>
> Thanks for your suggestion. We have evaluated our method on an NVIDIA RTX 3090 GPU and report inference times in Table 1 below. On average, our approach adds only 0.79 – 1.04 seconds per sample compared to the baseline control models. Compared to the training‑free layout‑to‑image control method LayoutGuidance, our method does not rely on backward guidance and thus avoids its substantial time cost. In contrast to existing training‑based approaches that require additional fine‑tuning on downstream datasets, our framework can be applied directly without introducing extra training overhead. Moreover, our method is fully compatible with structural controllable models such as ControlNet, yielding superior performance while incurring less than one additional second per sample.
>
> We also note that further acceleration is possible by integrating more efficient grounding models (for example, EfficientSAM [1]) or memory‑efficient attention libraries (such as xFormers [2]). We will incorporate these quantitative runtime results and discussions into the revised manuscript to clarify the practical efficiency of our approach.
>
>
> Table 1: Inference time comparison with existing methods on COCO dataset.
> | Method                         | Inference time (s)    |
> | ------------------------------ | --------------------- |
> | ControlNet                     | 2.55                  |
> | Layout Guidance                | 12.58                 |
> | Instance Diffusion             | 8.98                  |
> | ControlNet + ReCon             | 3.34 (2.55 + 0.79)    |
> | Instance Diffusion + ReCon     | 10.02 (8.98 + 1.04)   |
>
> **Re Weakness #2:**
>
> Thank you for the suggestion to perform an ablation study on the choice of rectification stages. We have evaluated rectification applied at individual timesteps as well as in combinations, and the results are reported in Table 2. Applying rectification only at the early stage (0.75 T) yields a modest improvement over the ControlNet baseline (mAP 35.2 vs. 34.9), but it is not optimal. We attribute this to two factors: first, early-stage rectifiaction can create shortcuts for spurious regions to rectify through later synthesis; second, features at 0.75 T remain highly diverse and partially undeveloped, so a single early rectifiaction cannot fully suppress all errors. When we combine rectification at multiple stages, we observe consistent gains: two stages rectifiaction ([0.5, 0.25 T]) raises mAP to 35.3, and using three stages ([0.5, 0.25, 0.1 T]) further increases mAP to 35.4. Our full four‐stage schedule ([0.75, 0.5, 0.25, 0.1 T]) achieves the best balance of all metrics (mAP 35.5, AP50 56.2, AP75 38.4). Extending to six stages yields only marginal benefit (mAP 35.5, AP75 38.5), suggesting that gains saturate beyond four rectifiactions. We will include these quantitative sensitivity results in the revised manuscript to demonstrate that our chosen schedule is both effective and well justified.
>
> Table 2: Ablation study of region rectification at different diffusion timesteps.
> | Method                                  | mAP  | AP50 | AP75 |
> |-----------------------------------------|------|------|------|
> | ControlNet                              | 34.9 | 55.5 | 37.7 |
> | 0.75 T                                  | 35.2 | 55.7 | 38.0 |
> | 0.5 T                                   | 35.3 | 56.0 | 38.0 |
> | 0.25 T                                  | 35.3 | 55.8 | 38.1 |
> | 0.1 T                                   | 35.2 | 55.8 | 38.0 |
> | [0.5, 0.25] T                           | 35.3 | 55.8 | 38.2 |
> | [0.5, 0.25, 0.1] T                      | 35.4 | 56.0 | 38.4 |
> | [0.75, 0.5, 0.25, 0.1] T                | 35.5 | 56.2 | 38.4 |
> | [0.75, 0.625, 0.5, 0.375, 0.25, 0.1] T  | 35.5 | 56.2 | 38.5 |
>
> **Re Weakness #3:**
>
> Thank you for this insightful suggestion. In Figure 5, our method removes the extraneous zebra that GLIGEN generates outside the specified bounding box (row 1) and the superfluous sheep outside the region of interest (row 3). In addition, it correctly restores the person that ControlNet fails to generate (row 2). To make these improvements clearer, we will expand the main text and the figure caption with detailed descriptions of each row and add overlay labels directly on the images to highlight the rectified regions.
>
> **Re Question #1:**
>
> Thank you for this constructive question. We have explored training with mixed real and synthetic data by augmenting our real dataset with varying proportions of synthetic samples, as shown in Figure 4. These experiments demonstrate how different mixing ratios affect detector performance. In future work, we plan to investigate more sophisticated strategies, such as adaptive re‑weighting of real and synthetic samples during training, to further improve robustness in mixed-data scenarios.
>
> **Re Question #2:**
>
> Thank you for this insightful question. While our current rectification mechanism does not explicitly model highly complex layouts or occlusions, it nevertheless demonstrates surprising robustness in challenging scenarios, as illustrated by the results in Figure 5 and Figure 6 of the manuscript. To handle more complex object layouts and occlusions in future work, we plan to explore a layer‑based generation strategy, in which objects are rendered in a defined depth order and refined progressively. This approach would allow the model to reason about occlusion relationships and ensure that foreground and background elements are composed correctly.
>
> **Re Question #3:**
>
> Thank you for this valuable question. Our experiments demonstrate that the proposed ReCon method significantly improves alignment between generated content and annotations. We therefore anticipate that our method can be extended to other vision tasks where content‑annotation consistency is critical, for example instance segmentation and visual grounding. In future work, we plan to conduct comprehensive evaluations on these tasks to validate and refine our approach.
>
> [1] Xiong, Yunyang, et al.  "Efficientsam: Leveraged masked image pretraining for efficient segment anything." In CVPR 2024.
>
> [2] Lefaudeux, Benjamin, et al. "xformers: A modular and hackable transformer modelling library." 2022.

---

### Decision · Program_Chairs · 2025-09-17

**Decision:**

Accept (spotlight)

**Comment:**

Summary:
This paper presents ReCon, a training-free augmentation framework that improves region controllability in diffusion-based data generation for object detection. When adding rectification and alignment modules during sampling, it achieves consistent gains across datasets and detectors.

Strengths:
The method is simple, modular, and requires no finetuning. It is validated thoroughly with diverse benchmarks and consistently improves both fidelity and detection performance.

Weaknesses:
Reviewers initially raised concerns about heuristic design choices, runtime overhead, and unclear ablations. The rebuttal provided timing analysis, clarified experiments, and added missing results, which largely resolved these issues.

Overall, this paper provides a practical and well-supported contribution. ACs recommend acceptance.